# IRAK2 directs stimulus-dependent nuclear export of inflammatory mRNAs

Hao Zhou[1†], Katarzyna Bulek[1,2†], Xiao Li[3†], Tomasz Herjan[1†], Minjia Yu[1,4], Wen Qian[1], Han Wang[1], Gao Zhou[3], Xing Chen[1], Hui Yang[1], Lingzi Hong[1], Junjie Zhao[1], Luke Qin[1], Koichi Fukuda[5], Annette Flotho[6], Ji Gao[7], Ashok Dongre[7], Julie A Carman[7], Zizhen Kang[1,8,9], Bing Su[8,9,10], Timothy S Kern[11,12], Jonathan D Smith[13], Thomas A Hamilton[1], Frauke Melchior[6], Paul L Fox[13], Xiaoxia Li[1]*

[1]Department of Immunology, Lerner Research Institute, Cleveland Clinic, Cleveland, United States; [2]Department of Immunology, Faculty of Biochemistry, Biophysics and Biotechnology, Jagiellonian University, Krakow, Poland; [3]Department of Genetics, Stanford University School of Medicine, Stanford, United States; [4]Department of Medicine, Mount Auburn Hospital, Harvard Medical School, Cambridge, United States; [5]Department of Molecular Cardiology, Lerner Research Institute, Cleveland Clinic, Cleveland, United States; [6]Zentrum für Molekulare Biologie der Universität Heidelberg, DKFZ-ZMBH Alliance, Heidelberg, Germany; [7]Discovery Biology, Bristol-Myers Squibb, Princeton, United States; [8]Shanghai Institute of Immunology, Shanghai Jiao Tong University School of Medicine, Shanghai, China; [9]Department of Immunobiology and Microbiology, Shanghai Jiao Tong University School of Medicine, Shanghai, China; [10]Department of Immunobiology, Vascular Biology and Therapeutics Program, Yale University School of Medicine, New Haven, United States; [11]School of Medicine, Case Western Reserve University, Cleveland, United States; [12]Stokes Veterans Administration Hospital, Cleveland, United States; [13]Department of Cellular and Molecular Medicine, Lerner Research Institute Cleveland Clinic, Cleveland, United States

*For correspondence:
lix@ccf.org

†These authors contributed equally to this work

Competing interests: The authors declare that no competing interests exist.

**Abstract** Expression of inflammatory genes is determined in part by post-transcriptional regulation of mRNA metabolism but how stimulus- and transcript-dependent nuclear export influence is poorly understood. Here, we report a novel pathway in which LPS/TLR4 engagement promotes nuclear localization of IRAK2 to facilitate nuclear export of a specific subset of inflammation-related mRNAs for translation in murine macrophages. IRAK2 kinase activity is required for LPS-induced RanBP2-mediated IRAK2 sumoylation and subsequent nuclear translocation. Array analysis showed that an SRSF1-binding motif is enriched in mRNAs dependent on IRAK2 for nuclear export. Nuclear IRAK2 phosphorylates SRSF1 to reduce its binding to target mRNAs, which promotes the RNA binding of the nuclear export adaptor ALYREF and nuclear export receptor Nxf1 loading for the export of the mRNAs. In summary, LPS activates a nuclear function of IRAK2 that facilitates the assembly of nuclear export machinery to export selected inflammatory mRNAs to the cytoplasm for translation.
DOI: https://doi.org/10.7554/eLife.29630.001

## Introduction

Eukaryotic cells produce mRNA in the nucleus through a series of events including 5' capping, 3'-end processing and splicing, which are coupled with transcription. Once these processes are

**eLife digest** The innate immune system is the body's first line of defense against invading microbes. Some immune cells carry specific receptor proteins called Toll-like receptors that can identify microbes and the signals they emit. As soon as the receptors have detected a threat – for example through sensing oily molecules that make up the cell membranes of microbes – they produce signaling proteins called cytokines and chemokines to alert other immune cells.

The DNA in the cell's nucleus carries the instructions needed to make proteins. To produce proteins, including cytokines and chemokines, the information first has to be transferred into mRNA templates, which carry the instructions to the sites in the cell where the proteins are made. Cytokine and chemokine mRNAs are generally short-lived, but previous studies in 2009 and 2011 have shown that an enzyme called IRAK2 can stabilize them to make them last longer. IRAK enzymes are activated by the Toll-like receptors after a threat has been detected. However, until now it was not known whether IRAK2 also helps to transport the mRNAs of cytokines and chemokines out of the cell nucleus.

Using immune cells of mice, Zhou et al. – including some of the researchers involved in the previous studies – discovered that IRAK2 helped to export the mRNAs of cytokines and chemokines from the immune cell nucleus into the surrounding cell fluid. The Toll-like receptors recognized the oily molecules of the microbes and consequently activated IRAK2, which lead to IRAK2 being moved into the cell nucleus.

Once activated, IRAK2 helped to assemble the export machinery that moved selected mRNAs out of the nucleus to build the proteins. To do so, IRAK2 stopped a destabilizing protein from binding to the mRNA, so that instead the export machinery could transport the mRNA of the cytokines and chemokines out of the cell nucleus.

A next step will be to test whether IRAK2 is required to guide exported mRNA tothe sites in the cell where the proteins are made. This new insight could help to develop new treatments for various diseases. For example, diseases in which the immune system attacks the cells of the body, rather than invaders, can be caused by too many cytokines and chemokines. Since IRAK2 helps to control the availability of cytokines and chemokines it may in future be used as a new drug target.
DOI: https://doi.org/10.7554/eLife.29630.002

complete, mRNA is exported from the nucleus to the cytoplasm where it can be translated to generate proteins. Nxf1 (also known as TAP) is the key mRNA export receptor, which only binds processed mRNA. Upon completion of mRNA processing, Nxf1 is recruited to the mRNA along with the TREX complex to promote the nuclear export the mRNA. The nuclear export adaptor ALYREF, a subunit of the TREX complex, plays a critical role in integrating the signals provided by mRNA processing and in triggering nuclear export receptor Nxf1 loading for the export of the target mRNAs (*Hung et al., 2010*; *Viphakone et al., 2012*; *Zhou et al., 2000*). ALYREF-Nxf1 interaction provides a mark on mRNA to signify that nuclear RNA-processing events are complete, and the processed mRNA is ready for export to the cytoplasm. Several other proteins have also been implicated as nuclear export adaptors, including the shuttling SR (serine- and arginine-rich) proteins 9G8, SRp20 and SRSF1 (*Huang et al., 2003*). However, it remains unclear how these nuclear export adaptors interact with different classes of mRNAs to achieve sequence-specific, stimulus-dependent export of target transcripts.

Toll-like receptor (TLR) signaling regulates the expression of chemokines and cytokines at both transcriptional and posttranscriptional levels (*Anderson, 2008*). Cytokine and chemokine mRNAs have short half-lives because of conserved cis-elements in their three prime untranslated regions (3' UTRs). Much effort has been devoted to understand how the conserved cis-elements within the 3' UTR can be recognized by RNA-binding proteins (including SRSF1) that function to mediate mRNA decay (*Fenger-Grøn et al., 2005*; *Lykke-Andersen and Wagner, 2005*; *Mayr, 2016*; *Sun et al., 2011*). Nevertheless, whether and how cytokine and chemokine mRNAs are regulated during nuclear export in response to TLR stimulation is unknown. TLRs transduce signals through the adaptor molecule MyD88 and IL-1R-associated kinase (IRAK) family members, including IRAK1, IRAK2 and IRAK4 (*Kawai and Akira, 2011*). IRAK4 is the upstream kinase of IRAK1 and IRAK2 (*Kim et al., 2007*;

Lin et al., 2010). IRAK1 is necessary for TAK1-dependent NFκB activation for the transcription of chemokines and cytokines (Cui et al., 2012; Yao et al., 2007). IRAK2 is an atypical kinase that mediates posttranscriptional regulation of inflammatory transcripts. Deletion of IRAK2 impairs the production of inflammatory cytokines and chemokines in macrophages in response to TLR stimulation (Wan et al., 2009; Yin et al., 2011) without affecting transcription of inflammatory genes. However, the precise mechanism by which IRAK2 controls the posttranscriptional regulation of inflammatory cytokine and chemokine production is an evolving area of investigation.

Here, we report that LPS induces IRAK2 nuclear localization to facilitate nuclear export of a subset of inflammatory mRNAs (including Cxcl1, Tnf and Cxcl2) to the cytosol for protein translation. IRAK2 kinase activity is required for LPS-induced RanBP2-mediated IRAK2 sumoylation, which facilitates IRAK2 nuclear translocation. By array analysis, we identified an SRSF1-binding motif enriched selectively in mRNA targets dependent on IRAK2 for nuclear export. IRAK2 phosphorylates SRSF1 and thereby reduces SRSF1 binding to the target mRNAs. On the other hand, SRSF1 knockdown resulted in increased nuclear export of the target mRNAs. Importantly, LPS induced the interaction of IRAK2 with the nuclear export factors ALYREF and Nxf1 recruiting them to the target mRNAs. While the depletion of SRSF1 allowed nuclear export adaptor ALYREF binding to the target mRNAs, LPS/IRAK2-induced recruitment of nuclear export receptor Nxf1 to the target transcripts was abolished by the knockdown of ALYREF. Thus, SRSF1-mediated nuclear sequestration of target mRNAs might be achieved by blocking the binding of ALYREF and Nxf1 to the mRNAs. Taken together, our results suggest that while SRSF1 binding renders target mRNAs sensitive for nuclear export, LPS promotes a nuclear function of IRAK2 that mediates the removal of SRSF1 and facilitates the assembly of nuclear export machinery to export the inflammatory mRNAs to the cytosol for protein translation.

## Results

### LPS induced IRAK2 modification and nuclear translocation

We and others have previously reported that IRAK2 plays a critical role in the production of pro-inflammatory cytokines and chemokines in response to TLR stimulation. However, the detailed molecular mechanism is not completely understood. IRAK2 is an atypical kinase due to the amino acid substitution of key catalytic residues (Asp -> Asn[333]; Asp ->His[351] Figure 1A), which is supported by the demonstration of catalytic activities of atypical kinases such as KSR2 (Brennan et al., 2011) and CASK (Mukherjee et al., 2008). The recombinant wild-type IRAK2 as well as IRAK2 mutants (ATP-binding site mutant KK235AA and catalytic site mutant H351A) were purified by using a bacterial expression system. While the purified recombinant wild-type IRAK2 was able to autophosphorylate and phosphorylate myelin basic protein (MBP), the IRAK2 mutants (KK235AA and H351A) displayed minimum activity (Figure 1B). Wild-type IRAK2, but not IRAK2 kinase-inactive mutants (ATP-binding site mutant KK235AA; catalytic site mutants H351A and N333A), restored LPS-mediated induction of inflammatory cytokines and chemokines in IRAK2-deficient bone marrow-derived macrophages (Figure 1C and E), suggesting that the kinase activity of IRAK2 is necessary for ligand-induced pro-inflammatory gene expression. To further investigate the role of IRAK2 in LPS-induced inflammatory gene expression, we recently generated IRAK2 kinase-inactive knockin mice in which the ATP-binding site was mutated (KK235AA, Figure 1—figure supplement 2). LPS-induced production of inflammatory cytokines and chemokines was greatly reduced in the macrophages from IRAK2 kinase-inactive knockin mice compared to that of the wild-type control mice (Figure 1D and F).

Notably, IRAK2 is modified in bone marrow-derived macrophages in response to LPS stimulation (Figure 1E). Wild-type IRAK2, but not IRAK2 kinase-inactive mutants (ATP-binding site mutant KK235AA; catalytic site mutants H351A and N333A), restored IRAK2 modification in IRAK2-deficient bone marrow-derived macrophages (Figure 1E). Interestingly, we found that the modified form of IRAK2 was translocated into the nucleus in a ligand-dependent manner (Figure 1F). Immunofluorescent staining showed that IRAK2 ATP-binding site mutant KK235AA and catalytic site mutants (H351A and N333A), failed to translocate into the nucleus, suggesting that IRAK2 kinase activity is important for its nuclear translocation (Figure 1G). In support of this, LPS-induced IRAK2 nuclear translocation was abolished in the macrophages from IRAK2 kinase-inactive knockin mice

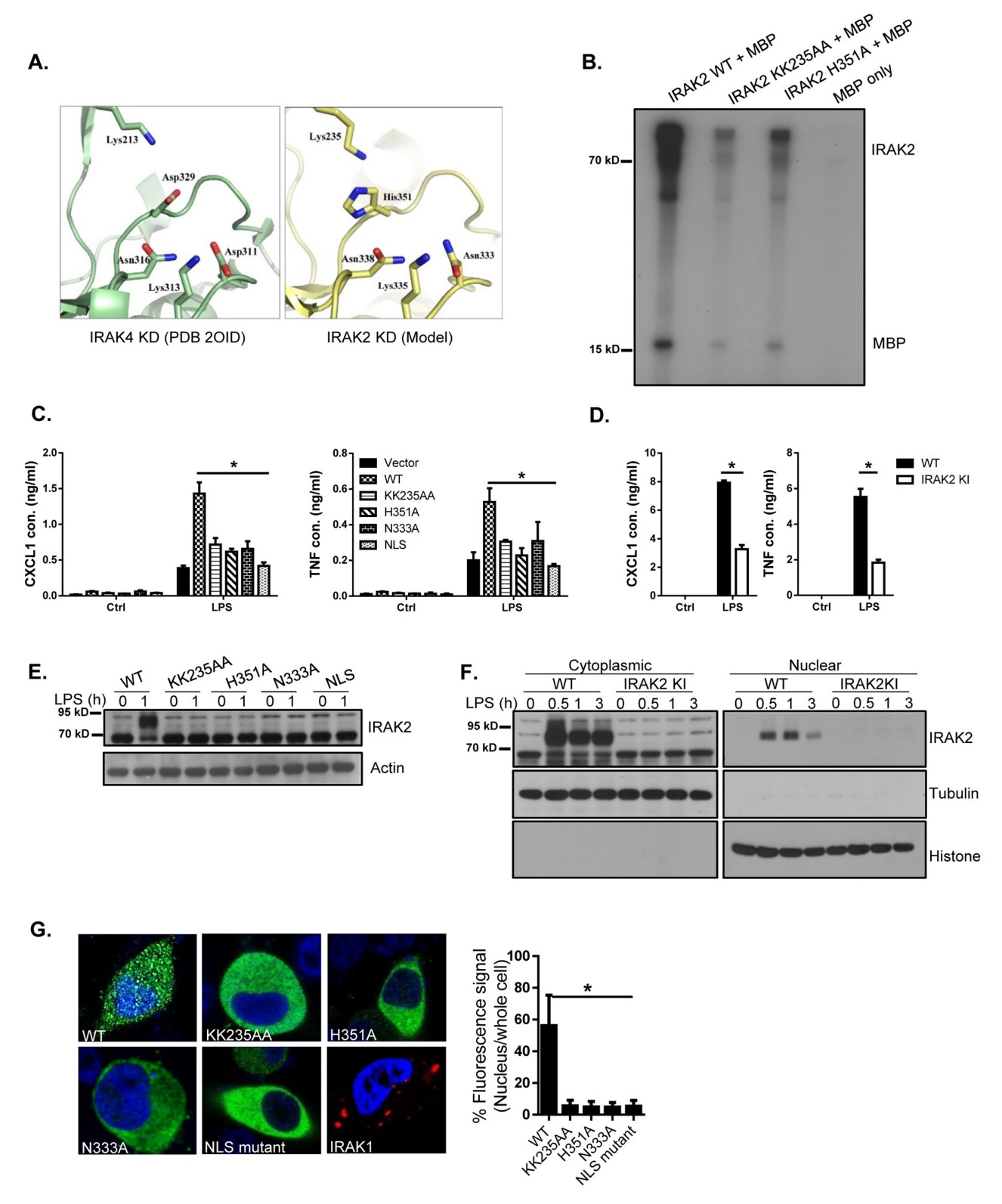

**Figure 1.** IRAK2 kinase activity is required for LPS-induced IRAK2 nuclear translocation. (**A**) A structural model of the catalytic site of IRAK2 kinase domain was created on a web-based molecular modeling server (*Arnold et al., 2006*), using the crystal structure of the IRAK4 KD segment (PDB 2OID) as a template (left). (**B**) In vitro kinase assay of purified recombinant mouse His-flag tagged IRAK2 WT, KK235AA and H351A mutants using myelin basic protein (MBP) as a substrate. (**C**) IRAK2-deficient macrophages restored with IRAK2 WT, KK235AA, H351A, N333A or empty vector were treated with

*Figure 1 continued on next page*

*Figure 1 continued*

LPS for 6 hr, followed by ELISA for CXCL1 and TNF levels in the supernatant. Western blot analysis for the expression of the transfected constructs are shown in (E-D) Wild-type (WT) and IRAK2 kinase-inactive (KI) BMDMs were treated with LPS for 6 hr, followed by ELISA for CXCL1 and TNF levels in the supernatant. (E) Lysates made from cells described in (C) were analyzed by western blot analysis with the indicated antibodies. (F) Cytoplasmic and nuclear extracts from WT and IRAK2 KI BMDMs treated with LPS for the indicated times were analyzed by western blot analysis with the indicated antibodies. The experiments were repeated five times with similar results. (G) Confocal imaging of HeLa cells transiently transfected with FLAG-tagged IRAK2 WT, KK235AA, H351A, N333A, NLS mutant and IRAK1-RFP (a negative control). For Flag-tagged construct, immunostaining was performed with anti-FLAG (green) on the transfected cells. Nuclei were stained with DAPI in blue. Fluorescence intensity was quantified in the nucleus versus the whole cell of 50 expressing cells and analyzed by Student's t test. Bar graph shows the percentage of fluorescent signal in the nucleus in expressing cells. Data represent mean ± SEM; *p<0.05 by Student's t test.
DOI: https://doi.org/10.7554/eLife.29630.003

The following source data and figure supplements are available for figure 1:

**Source data 1.** The numerical data for the graphs in *Figure 1*.
DOI: https://doi.org/10.7554/eLife.29630.008

**Figure supplement 1.** Coomassie blue staining for recombinant proteins using in *Figure 1*.
DOI: https://doi.org/10.7554/eLife.29630.004

**Figure supplement 2.** Sequencing analysis of wild-type, IRAK2 kinase-inactive (IRAK2 KI) and heterozygous mice.
DOI: https://doi.org/10.7554/eLife.29630.005

**Figure supplement 3.** Lower magnification of the confocal imaging of *Figure 1G*.
DOI: https://doi.org/10.7554/eLife.29630.006

**Figure supplement 4.** Cytoplasmic and nuclear extracts from WT BMDMs treated with oxidized low-density lipoprotein (ox-LDL), R848 and CpGB for the indicated times were analyzed by western blot analysis with the indicated antibodies.
DOI: https://doi.org/10.7554/eLife.29630.007

(*Figure 1F*). Interestingly, not only LPS stimulation leads to IRAK2 modification and nuclear localization, endogenous TLR4 ligand oxidized low-density lipoprotein (ox-LDL), TLR7 ligand R848 and TLR9 ligand CpGB stimulation can also induce IRAK2 modification and subsequent nuclear translocation (*Figure 1—figure supplement 4*).

Nuclear localization signal (NLS) is a short peptide motif that mediates the nuclear import of proteins by binding to importin complex. We identified a putative NLS in the IRAK2 kinase domain (Aa 357–366: HPDNKKTKYT), which was mutated by substituting the three lysine residues (K361, K362 and K364) with alanines. Immunofluorescent staining showed that the IRAK2 NLS mutant indeed failed to translocate into the nucleus (*Figure 1G* and *Figure 1—figure supplement 3*). IRAK2 NLS mutant also failed to restore LPS-induced inflammatory cytokine and chemokine production in IRAK2-deficient macrophages (*Figure 1C and E*), indicating that IRAK2 nuclear translocation is required for IRAK2-mediated inflammatory gene expression. While we showed that modified IRAK2 was translocated into the nucleus (*Figure 1F*), IRAK2 NLS mutant failed to be modified (*Figure 1E*), implicating a critical link between IRAK2 nuclear translocation and modification.

## RanBP2-mediated IRAK2 sumoylation is required for its nuclear translocation

We then investigated how LPS induced IRAK2 nuclear translocation. Mass spectrometry (MS) analysis showed that IRAK2 interacts with importin-β (*Figure 2A* and *Figure 2—figure supplement 1*), and co-immunoprecipitation experiment implicates the formation of IRAK2-RanBP2 complex (*Figure 2A*). While importin-β is one of the transport receptor mediating translocation of molecules into the cell nucleus, RanBP2 is a major nucleoporin that extends cytoplasmic filaments from the nuclear pore complex and contains phenylalanine–glycine repeats that bind nuclear transport receptors of the importin beta family (*Wu et al., 1995*; *Yokoyama et al., 1995*). Interestingly, we found that IRAK2 ATP-binding site mutant KK235AA and catalytic site mutants H351A/N333A retained the interaction with importin-β, whereas NLS mutant failed to form a complex with importin-β (*Figure 2A*). These results suggest that the IRAK2 kinase activity is not required for IRAK2's NLS to engage the importin complex. On the other hand, IRAK2 KK235AA and H351A/N333A kinase-inactive mutants and as well as NLS mutant lost interaction with RanBP2 (*Figure 2A*). It has been reported that LPS stimulation induces the phosphorylation of IRAK2 at S136 and T140 (*Weintz et al., 2010*). We found that mutation at these two phosphorylation sites (IRAK2 S136A/

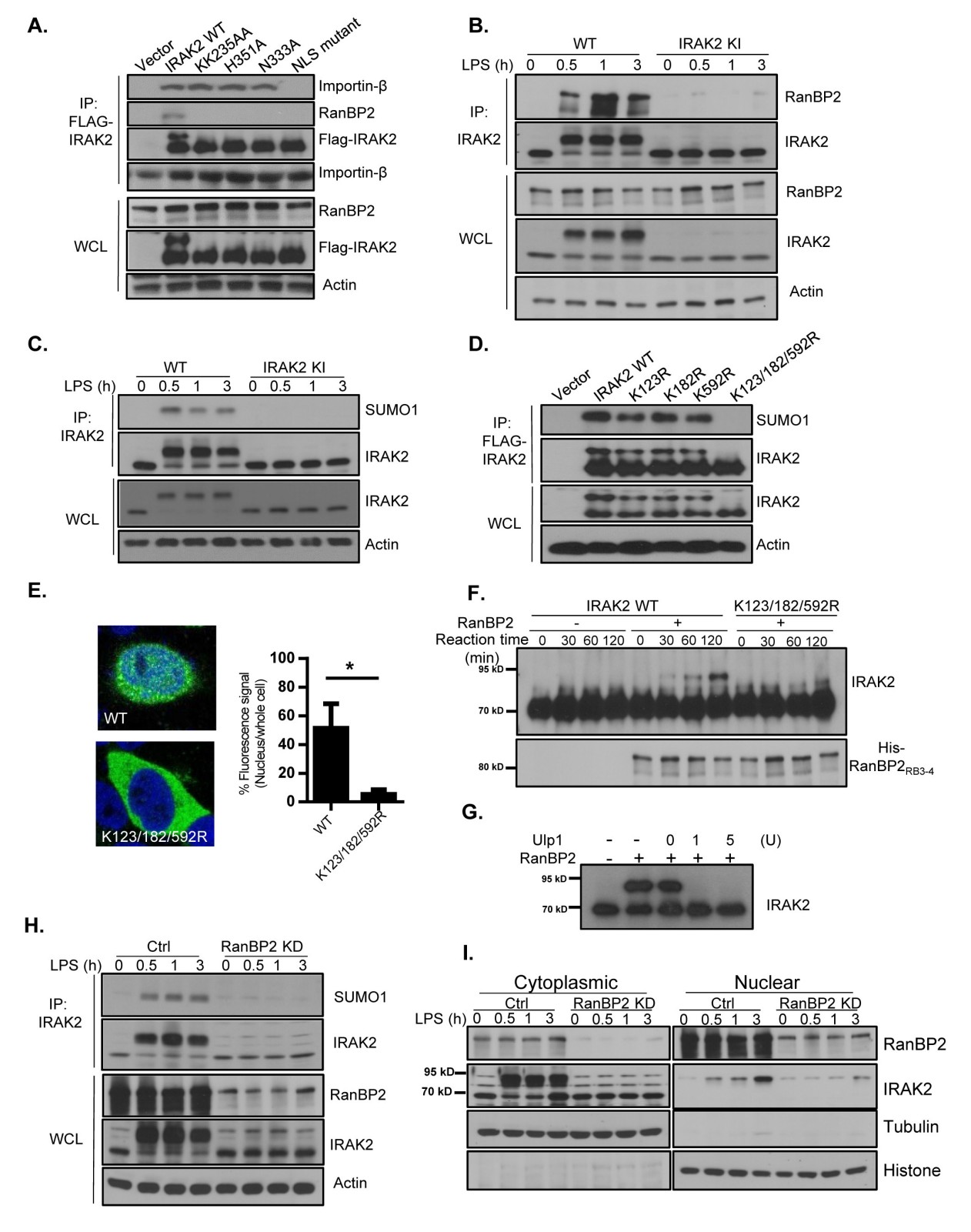

**Figure 2.** RanBP2 mediates LPS-induced IRAK2 sumoylation. (**A**) HeLa cells transiently transfected with FLAG-tagged IRAK2 WT, KK235AA, H351A, N333A and NLS mutant (K361/362/364A) followed by immunoprecipitation (IP) with anti-FLAG antibody and analyzed by western blot analysis with the indicated antibodies. (**B**) Wild-type (WT) and IRAK2 kinase inactive (KI) BMDMs were treated with LPS for indicated times, followed by immunoprecipitation (IP) with anti-IRAK2 antibody and analyzed by western blot analysis with the indicated antibodies. (**C**) WT and IRAK2 KI BMDMs
*Figure 2 continued on next page*

*Figure 2 continued*

were treated with LPS for the indicated times, followed by immunoprecipitation (IP) with anti-IRAK2 antibody under the denaturing condition (0.1% SDS in the lysis buffer) and analyzed by western blot analysis with the indicated antibodies. (D) HeLa cells transiently transfected with FLAG-tagged IRAK2 WT, K123R, K182R, K592R and K123/182/592R mutant followed by immunoprecipitation (IP) with anti-FLAG antibody under the denaturing condition (0.1% SDS in the lysis buffer) and analyzed by western blot analysis with the indicated antibodies. (E) Confocal imaging of HeLa cells transiently transfected with FLAG-tagged IRAK2 WT and K123/182/592R. Immunostaining was performed with anti-FLAG (green) on the transfected cells. Nuclei were stained with DAPI in blue. Fluorescence intensity was quantified in the nucleus versus the whole cell of 50 expressing cells and analyzed by Student's t test. Bar graph shows the percentage of fluorescent signal in the nucleus in expressing cells. Data represent mean ± SD; *p<0.05. (F–G) In vitro sumoylation assay using purified recombinant Aos1/Uba2 (E1), UBC9 (E2), RanBP2$_{RB3-4}$ (E3), SUMO1 and IRAK2 WT and IRAK2 K123/182/592R, following by western blot analysis with anti-IRAK2 antibody. RanBP2 was detected by western blot prior to the reaction using anti-His antibody. IRAK2 was sumoylated in vitro, followed by incubation with Ulp1 (1 unit and 5 unit) for 30 min (G). (H) Control (non-targeting siRNA) and RanBP2 knockdown (KD, with RanBP2 siRNA) macrophages were treated with LPS for the indicated times, followed by immunoprecipitation (IP) with anti-IRAK2 antibody and analyzed by western blot analysis with the indicated antibodies.(I) Cytoplasmic and nuclear extracts from Control (non-targeting siRNA) and RanBP2 knockdown (KD, with RanBP2 siRNA) treated with LPS for indicated times were analyzed by western blot analysis with the indicated antibodies. The experiments were repeated for five times with similar results. Data represent mean ± SEM; *p<0.05 by Student's t test.

DOI: https://doi.org/10.7554/eLife.29630.009

The following source data and figure supplements are available for figure 2:

**Source data 1.** The numerical data for the graphs in *Figure 2E*.
DOI: https://doi.org/10.7554/eLife.29630.017

**Figure supplement 1.** IL1R-293 cells were transiently transfected with HA-tagged IRAK2 and treated by IL-1.
DOI: https://doi.org/10.7554/eLife.29630.010

**Figure supplement 1—source data 1.** The numerical data for the graphs in *Figure 2—figure supplement 2*.
DOI: https://doi.org/10.7554/eLife.29630.011

**Figure supplement 2.** The phosphorylation of IRAK2 is required for its nuclear translocation.
DOI: https://doi.org/10.7554/eLife.29630.012

**Figure supplement 3.** RanBP2 mediates IRAK2 nuclear translocation.
DOI: https://doi.org/10.7554/eLife.29630.013

**Figure supplement 4.** Lower magnification of the confocal imaging in showing in *Figure 2E*.
DOI: https://doi.org/10.7554/eLife.29630.014

**Figure supplement 5.** Coomassie blue staining of purified recombinant Aos1/Uba2 (E1), UBC9 (E2), RanBP2 (E3) and SUMO1.
DOI: https://doi.org/10.7554/eLife.29630.015

**Figure supplement 6.** SUMO protease Ulp1 removes IRAK2 modification.
DOI: https://doi.org/10.7554/eLife.29630.016

T140A double mutant) abolished LPS-induced interaction of IRAK2 with RanBP2 and its nuclear localization (*Figure 2—figure supplement 2*), although the IRAK2 S136A/T140A double mutant still retained the interaction with importin-β. These findings suggest that the activation of IRAK2 may result in auto-phosphorylation at S136A and T140A, which in turn mediates the interaction with RanBP2. In support of this, it is indeed the modified form of IRAK2 was specifically co-immunoprecipitated by RanBP2 (*Figure 2—figure supplement 3*). Furthermore, the phos-tag gel electrophoresis confirmed that the phosphorylated IRAK2 preferentially binds to RanBp2 (*Figure 2—figure supplement 3*).

Notably, the C-terminal of RanBP2 carries an active SUMO E3 ligase domain, which was shown to modulate cytoplasmic and nuclear transport of macromolecules (*Pichler et al., 2002*). The modified IRAK2 band induced by LPS stimulation was about 20 kDa bigger than the unmodified IRAK2. Sumoylation typically gives rise to a 20 KD size shift in the SDS-PAGE gel, even though it is only 11 KD in molecular mass (*Werner et al., 2012*). Importantly, IRAK2 is characterized as a sumoylated protein in recent proteomics study (*Lamoliatte et al., 2014*). Thus, we hypothesized that IRAK2 is sumoylated upon LPS stimulation via its interaction with RanBP2, which might play an important role in IRAK2 nuclear translocation. LPS stimulation indeed induced the interaction between IRAK2 and RanBP2 (*Figure 2B*). We found LPS-induced IRAK2-RanBP2 interaction was abolished in the macrophages from IRAK2 kinase-inactive knockin mice (*Figure 2B*), confirming that the kinase activity of IRAK2 is required for its interaction with RanBP2. Furthermore, SUMO1 was detected in IRAK2 immunoprecipitates under denaturing condition, these results indicate that IRAK2 is conjugated by SUMO1 in response to LPS stimulation (*Figure 2C*). Consistent with the fact that IRAK2 kinase-inactive mutant failed to interact with RanBP2, LPS-induced IRAK2 sumoylation was abolished in the

macrophages from IRAK2 kinase-inactive knockin mice (*Figure 2C*). To identify the critical lysine residues required for sumoylation, the lysine residues in the sumoylation consensus sequence (ψKxE) were mutated to arginines. IRAK2 K123/182/592R triple mutant lost sumoylation, and failed to translocate into the nucleus (*Figure 2D–E* and *Figure 2—figure supplement 4*), suggesting that IRAK2 sumoylation plays an essential role for IRAK2 nuclear translocation.

To determine whether RanBP2 is the E3 ligase for IRAK2, we performed in vitro sumoylation assay using purified recombinant Aos1/Uba2 (E1), UBC9 (E2), RanBP2 (E3), and SUMO1. By in vitro sumoylation assay, we found that wild-type IRAK2 was sumoylated, whereas the K123/182/592R triple mutant failed to be modified (*Figure 2F* and *Figure 2—figure supplement 5*). The modified form of IRAK2 disappeared after treatment with SUMO-specific isopeptidase Ulp1, confirming the observed modification of IRAK2 was indeed due to sumoylation (*Figure 2G*). Interestingly, the interaction between IRAK2 and RanBP2/SUMO1 was also sensitive to Ulp1 treatment (*Figure 2—figure supplement 6*). To further assess the importance of RanBP2 in LPS-induced IRAK2 modification, RanBP2 was knocked down in macrophages. LPS-induced IRAK2 sumoylation was abolished in the absence of RanBP2, confirming that RanBP2 is the E3 ligase for IRAK2 sumoylation (*Figure 2H*). Co-immunoprecipitation experiment showed that the sumoylated form of IRAK2 preferentially binds to RanBP2 (*Figure 2—figure supplement 3*). Furthermore, LPS-induced IRAK2 nuclear translocation was abolished in RanBP2 knock-down cells, supporting the critical role of IRAK2 sumoylation for its nuclear translocation (*Figure 2I* and *Figure 2—figure supplement 3*).

## IRAK2 mediates nuclear export of mRNAs of inflammatory genes

We next investigated the functional importance of IRAK2 in the nucleus. Notably, IRAK2 is required for the posttranscriptional regulation of inflammatory genes in response to LPS stimulation (*Figure 3—figure supplement 1*; [*Wan et al., 2009*]). The subdivision of eukaryotic cells into nuclear and cytoplasmic compartments enables spatial separation of transcription and translation. We hypothesized that LPS-induced IRAK2 nuclear localization may facilitate nuclear export of the inflammatory mRNAs to cytosol for translation after their transcription in the nucleus. To test this hypothesis, we isolated nuclear and cytoplasmic RNA from untreated and LPS-treated wild-type and IRAK2-deficient macrophages, followed by array analysis. We identified a group of mRNAs (including *Cxcl1*, *Tnf* and *Cxcl2*) accumulated in the nucleus in the absence of IRAK2, suggesting that IRAK2 is required for the exportation of these mRNAs (*Figure 3A* and *Figure 3—figure supplement 2*). The same mRNAs (including *Cxcl1*, *Tnf* and *Cxcl2*) were also accumulated in the nucleus of IRAK2 kinase inactive macrophages, suggesting that the kinase activity of IRAK2 is required for the exportation of these mRNAs (*Figure 3B*). To identify the potential role of RNA-binding proteins in IRAK2-mediated regulation of mRNA nuclear export, we applied RNA-READ motif scanner, a regression-based framework which searches for previously defined RNA cis-motifs (*Ray et al., 2013*) whose involvement in the regression significantly improves the fitting to the data than ones based on the background distribution of the sequence alone (Materials and methods). As a result, we found that the binding sites of SRSF1 (KGRWGSM, K: G/U; R:G/A; W:A/U; S:G/C; M:A/C) are significantly enriched in the 3'UTRs of the positive set (LPS-induced mRNAs that were accumulated in the nucleus of IRAK2-deficient macrophages compared to that in wild-type macrophages) versus the negative set (LPS-induced mRNAs whose cytosol/nuclear distribution in the macrophages was not affected by IRAK2 deficiency), and this motif significantly improves the fitting to the data than ones based on the background distribution (Likelihood ratio test p-value<0.0004) (*Figure 3C*).

SRSF1, also known as SF2/ASF is a serine- and arginine-rich protein, which plays important roles in mRNA metabolism. We hypothesized that the impact of IRAK2 on the nuclear exportation of the mRNAs (enriched with SRSF1-binding motifs) might be through IRAK2-mediated modulation on SRSF1. Interestingly, we detected SRSF1-IRAK2 interaction in the nucleus by duo-link assay (*Figure 3D* and *Figure 3—figure supplement 3*), which was abolished for IRAK2 kinase-inactive and sumoylation mutants. Furthermore, IRAK2 was able to phosphorylate SRSF1 in the in vitro kinase assay (*Figure 3E*). In support of this, we found that LPS indeed induced SRSF1 serine/threonine phosphorylation, which was diminished in IRAK2-kinase-inactive nucleus (*Figure 3F*). Previous studies have shown that SRSF1 binds RNA in its hypo-phosphorylated form (*Sanford et al., 2005*; *Xiao and Manley, 1997*), suggesting that IRAK2-mediated SRSF1 phosphorylation probably drives the dissociation of SRSF1 from the mRNA targets, promoting their nuclear export. By RNA immunoprecipitation, we indeed found that SRSF1-bound transcripts (*Cxcl1* and *Tnf*) were increased in IRAK2-

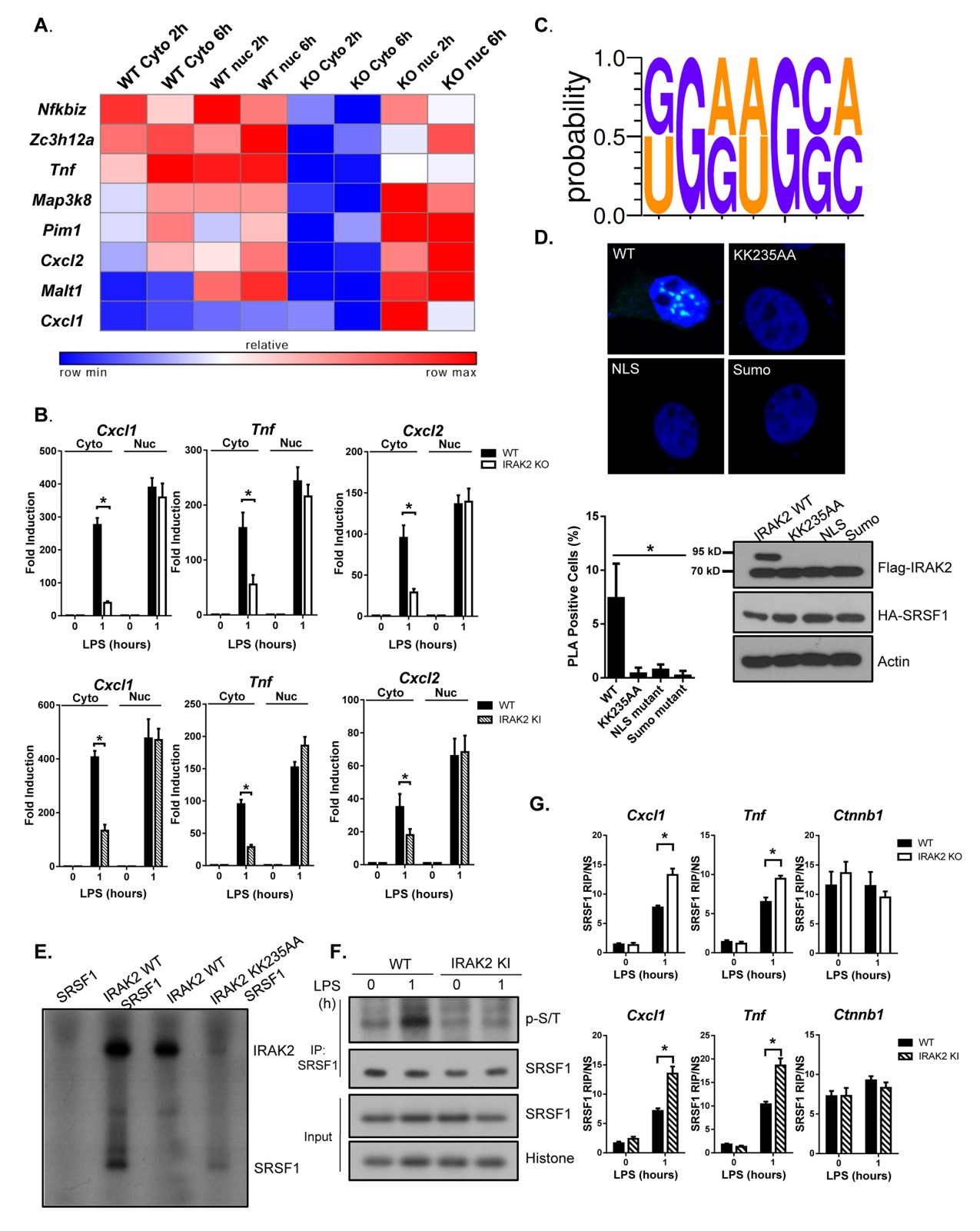

**Figure 3.** IRAK2 mediates nuclear export of mRNAs of pro-inflammatory genes. (**A**) Heatmap showing the cytosolic and nuclear abundance of selected transcripts that require IRAK2 for nuclear export in response to LPS stimulation. Affymetrix microarray was performed to analyze the total mRNAs isolated from the cytoplasmic and nuclear fractions of 3 pairs of WT and IRAK2 knockout (KO) BMDMs treated with LPS for the indicated times. Transcripts-induced by LPS in the nuclei of both WT and KO cells were further ranked based on an index (I) that measures the impact of IRAK2

*Figure 3 continued on next page*

*Figure 3 continued*

deficiency on transcript nuclear retention (See Materials and methods section). Scaled expression levels of transcripts encoding pro-inflammatory cytokines and chemokines are shown from blue to red, indicating low to high expression. (**B**) Total mRNAs isolated from the cytoplasmic and nuclear fractions of WT, IRAK2 KO and IRAK2 kinase-inactive (KI) BMDMs treated with LPS for the indicated times were subjected to RT-PCR analyses. (**C**) RNA-READ motif scanner, a regression-based framework which searches for previously defined RNA cis-motif was performed and the binding sites of SRSF1 (KGRWGSM, K: G/U; R:G/A; W:A/U; S:G/C; M:A/C) are significantly enriched in the 3'UTRs of the positive set versus the negative set (Materials and methods). (**D**) Confocal imaging of PLA (proximity ligation assay) signal of Hela cells transfected with HA-tagged SRSF1 together with FLAG-tagged IRAK2 WT, KK235AA, NLS mutant (K361/362/364A) and Sumo mutant (K123/182/592R). Mouse anti-HA and Rabbit anti-FLAG antibody were used for the proximity ligation assay. Green dots present PLA positive signal indicating the interaction of IRAK2 with SRSF1. Bar graph shows the percentage of PLA positive cells analyzed by Student's t test. *p<0.05. Western blot analysis of Hela cells transfected with HA-tagged SRSF1 together with FLAG-tagged IRAK2 WT, KK235AA, NLS mutant (K361/362/364A) and Sumo mutant (K123/182/592R) with the indicated antibody. (**E**) Phosphorylation of SRSF1 by IRAK2 was assessed by in vitro kinase assay using recombinant IRAK2 and SRSF1. (**F**) WT and IRAK2 KI BMDMs treated with LPS for the indicated times, followed by immunoprecipitation (IP) with anti-SRSF1 antibody under denaturing condition (0.1% SDS in the lysis buffer) and analyzed by western blot analysis with the indicated antibodies. (**G**) WT, IRAK2 KO and IRAK2 KI BMDMs were treated with LPS for indicated times, followed by RNA immunoprecipitation with anti-SRSF1 antibody and RT-PCR analyses of the indicated mRNAs. The presented are the relative values normalized against IgG control (Materials and methods). The experiments were repeated for five times with similar results. Data represent mean ± SEM; *p<0.05 by Student's t test.

DOI: https://doi.org/10.7554/eLife.29630.018

The following source data and figure supplements are available for figure 3:

**Source data 1.** The numerical data for the graphs in *Figure 3*.
DOI: https://doi.org/10.7554/eLife.29630.024

**Figure supplement 1.** IRAK2 is required for the posttranscriptional regulation of inflammatory genes in response to LPS stimulation.
DOI: https://doi.org/10.7554/eLife.29630.019

**Figure supplement 1—source data 1.** The numerical data for the graphs in *Figure 3—figure supplement 3*.
DOI: https://doi.org/10.7554/eLife.29630.020

**Figure supplement 2.** Heatmap showing the cytosolic and nuclear abundance of selected transcripts that require IRAK2 for nuclear export in response to LPS stimulation.
DOI: https://doi.org/10.7554/eLife.29630.021

**Figure supplement 3.** Additional Images for the proximity ligation assay showed in *Figure 3D*.
DOI: https://doi.org/10.7554/eLife.29630.022

**Figure supplement 4.** WT and IRAK2 KO BMDMs were treated with LPS for indicated times and fixed in 0.1% formaldehyde for 15 min at room temperature, whereupon the cross-linking reaction was stopped with glycine (pH 7; 0.25 M).
DOI: https://doi.org/10.7554/eLife.29630.023

deficient and IRAK2-kinase-inactive macrophages compared to that in wild-type cells (*Figure 3G* and *Figure 3—figure supplement 4*). Taken together, these results suggest that IRAK2 may directly phosphorylate SRSF1 in the nucleus in response to LPS stimulation, which facilitates the dissociation of SRSF1 from the LPS-induced transcripts, and the subsequent nuclear export of these mRNAs. Notably, SRSF1 also plays a critical role in regulating genes that are constitutively expressed, such as *Ctnnb1* (β-catenin) (*Fu et al., 2013*). Interestingly, SRSF1-bound transcript *Ctnnb1* was not affected by either LPS stimulation or IRAK2 deficiency (*Figure 3G*), indicating that IRAK2 may only regulate a specific subset of SRSF1-targeted mRNAs.

## IRAK2 mediates the recruitment of ALYREF nuclear export factor to the target mRNAs

The next question is how IRAK2 mediates nuclear export of the LPS-induced mRNAs bound by SRSF1. Nuclear export of the mature mRNA is an active process that is integrated with many of the nuclear processing steps, facilitated by RNA-binding proteins and transport factors (*Wickramasinghe and Laskey, 2015*). Importantly, ALYREF nuclear export factor was actually among the proteins based on MS analysis of IRAK2 immunoprecipitation (*Figure 2—figure supplement 1*). By coimmunoprecipitation, we found that LPS stimulation induced the interaction of IRAK2 with ALYREF in the nucleus (*Figure 4A*). Duo-link assay showed that ALYREF interacts with wild-type IRAK2, but not kinase-inactive and sumoylation mutants (*Figure 4B* and *Figure 4—figure supplement 1*). These results indicate that the kinase activity of IRAK2 and subsequent sumoylation are required for IRAK2's interaction with ALYREF in the nucleus, which is probably due to the fact that the kinase-inactive and sumoylation mutants failed to translocate into the nucleus (*Figure 1G*).

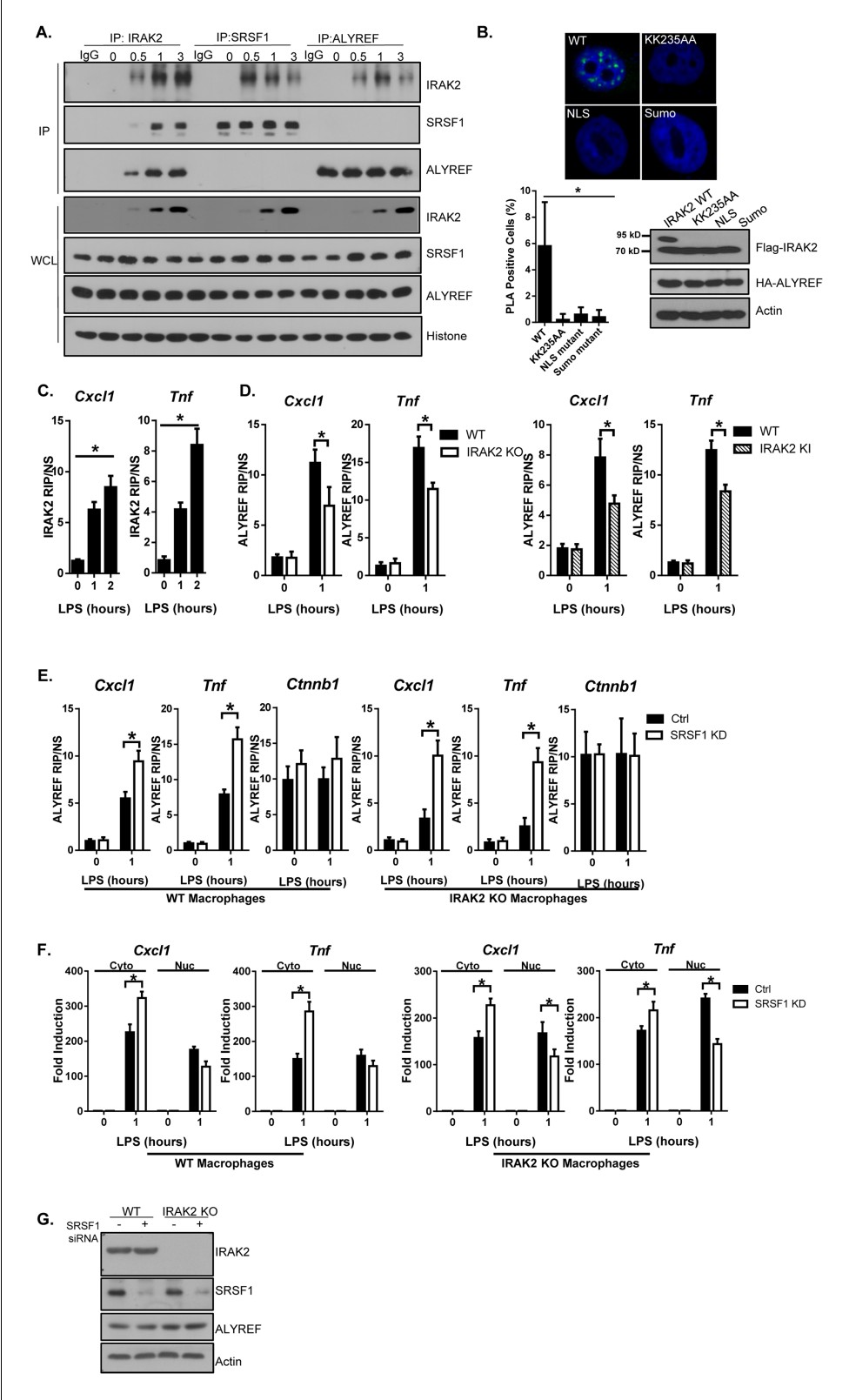

**Figure 4.** IRAK2 mediates the recruitment of ALYREF nuclear export factor to the target mRNAs. (**A**) Nuclear extracts were isolated from Wild-type (WT) BMDMs treated with LPS for the indicated times, followed by immunoprecipitation (IP) with anti-IRAK2, anti-SRSF1 or anti-ALYREF and analyzed by western blot analysis with the indicated antibodies. (**B**) Confocal imaging of PLA (proximity ligation assay) signal of Hela cells transfected with FLAG-tagged IRAK2 WT, KK235AA, NLS mutant (K361/362/364A) and Sumo mutant (K123/182/592R). Mouse anti-ALYREF and Rabbit anti-FLAG antibody

*Figure 4 continued on next page*

Figure 4 continued

were used for the proximity ligation assay. Green dots present PLA positive signal indicating the interaction of IRAK2 with ALYREF. Bar graph shows the percentage of PLA positive cells analyzed by Student's t test. *p<0.05. Western blot analysis of Hela cells transfected with FLAG-tagged IRAK2 WT, KK235AA, NLS mutant (K361/362/364A) and Sumo mutant (K123/182/592R) with the indicated antibody. (C) WT BMDMs were treated with LPS for the indicated times, followed by RNA immunoprecipitation with anti-IRAK2 antibody and RT-PCR analyses of the indicated mRNAs. The presented are the relative values normalized against IgG control (Materials and methods). (D) WT, IRAK2 KO and IRAK2 KI BMDMs were treated with LPS for the indicated times, followed by RNA immunoprecipitation with anti-ALYREF antibody and RT-PCR analyses of the indicated mRNAs. The presented are the relative values normalized against IgG control (Materials and methods). (E–G) SRSF1 was knocked down in WT and IRAK2 KO macrophages. The SRSF1-knockdown cells were then treated with LPS for the indicated times, followed by RNA immunoprecipitation with anti-ALYREF and RT-PCR analyses of the indicated mRNAs. The presented are the relative values normalized against IgG control (Materials and methods) (E). Total mRNAs were isolated from the cytoplasmic and nuclear fractions of The SRSF1-knockdown cells treated with LPS for the indicated times, followed by RT-PCR analyses for the indicated mRNAs (F). The experiments were repeated for at least three times. Data represent mean ± SEM; *p<0.05 by Student's t test. (G) Western analysis of lysates from cells used in *Figure 4E–F* with the indicated antibodies.

DOI: https://doi.org/10.7554/eLife.29630.025

The following source data and figure supplements are available for figure 4:

**Source data 1.** The numerical data for the graphs in *Figure 4*.

DOI: https://doi.org/10.7554/eLife.29630.029

**Figure supplement 1.** Additional images for the proximity ligation assay showed in *Figure 4B*.

DOI: https://doi.org/10.7554/eLife.29630.026

**Figure supplement 2.** IRAK2 is requried for the recruitment of ALYREF to the target mRNAs.

DOI: https://doi.org/10.7554/eLife.29630.027

**Figure supplement 2—source data 1.** The numerical data for the graph in *Figure 4—figure supplement 2*.

DOI: https://doi.org/10.7554/eLife.29630.028

Since LPS-induced the interaction of IRAK2 with SRSF1 (*Figures 3D–F* and *4A*), we wondered whether IRAK2-SRSF1 and IRAK2-ALYREF belong to one complex. Interestingly, ALYREF was absent in the SRSF1 immunoprecipitates, while SRSF1 was also not detected in ALYREF immunoprecipitates (*Figure 4A*). On the other hand, IRAK2 was detected in either SRSF1 or ALYREF immunoprecipitates in response to LPS stimulation (*Figure 4A*). These results suggest that LPS-induced IRAK2's interaction with SRSF1 and ALYREF are mutually exclusive. The fact that IRAK2 was able to phosphorylate SRSF1 (*Figure 3F*) led us hypothesize that IRAK2-mediated SRSF1 phosphorylation might drive SRSF1 off the mRNA targets, allowing ALYREF's binding to the target mRNAs. In support of this, RIP analysis showed that LPS stimulation induced the binding of IRAK2 and ALYREF to the mRNAs of *Cxcl1*, *Cxcl2* and *Tnf*, and ALYREF's binding to the mRNAs was substantially reduced in IRAK2-deficient and IRAK2-kinase-inactive macrophages (*Figure 4C–D*). Interestingly, ALYREF bound *Cxcl1* and *Tnf* transcripts were increased in SRSF1 knock-down cells, even in IRAK2 KO macrophages, indicating that ALYREF's binding to *Cxcl1* and *Tnf* is no longer IRAK2-dependent in the absence of SRSF1 (*Figure 4E and G* and *Figure 4—figure supplement 2*). Similar to SRSF1, ALYREF also binds to constitutively expressed *Ctnnb1* mRNA. Importantly, ALYREF-bound *Ctnnb1* mRNA was not affected by LPS stimulation, IRAK2 deficiency or SRSF1 knockdown (*Figure 4E*). These results confirmed that the role of IRAK2 for ALYREF's binding to the target mRNAs is probably to remove SRSF1 from the targets. Consistently, whereas IRAK2 deficiency resulted in nuclear accumulation of *Cxcl1* and *Tnf* mRNA, *Cxcl1* and *Tnf* mRNA was efficiently exported to the cytosol in SRSF1 knock-down cells in the absence of IRAK2 (*Figure 4F–G*).

## IRAK2-ALYREF facilitates the assembly of nuclear export complex on target mRNAs

The metazoan TREX complex is recruited to mRNA during nuclear RNA processing and functions in exporting mRNA to the cytoplasm (*Cheng et al., 2006*). Recent studies suggest that the recruitment of ALYREF to TREX allows for mRNA export by driving nuclear export receptor Nxf1 (also known as TAP) into a conformation capable of binding mRNA (*Viphakone et al., 2012*). In addition to the LPS-induced interaction of IRAK2 with ALYREF, we found that LPS stimulation also induced the interaction of IRAK2 with Nxf1 (*Figure 5A*). IRAK2 failed to interact with Nxf1 in the absence of ALYREF, suggesting that IRAK2-ALYREF interaction is required for ALYREF to facilitate the assembly of nuclear export complex to the target mRNAs, including the recruitment of Nxf1 (*Figure 5A*). In

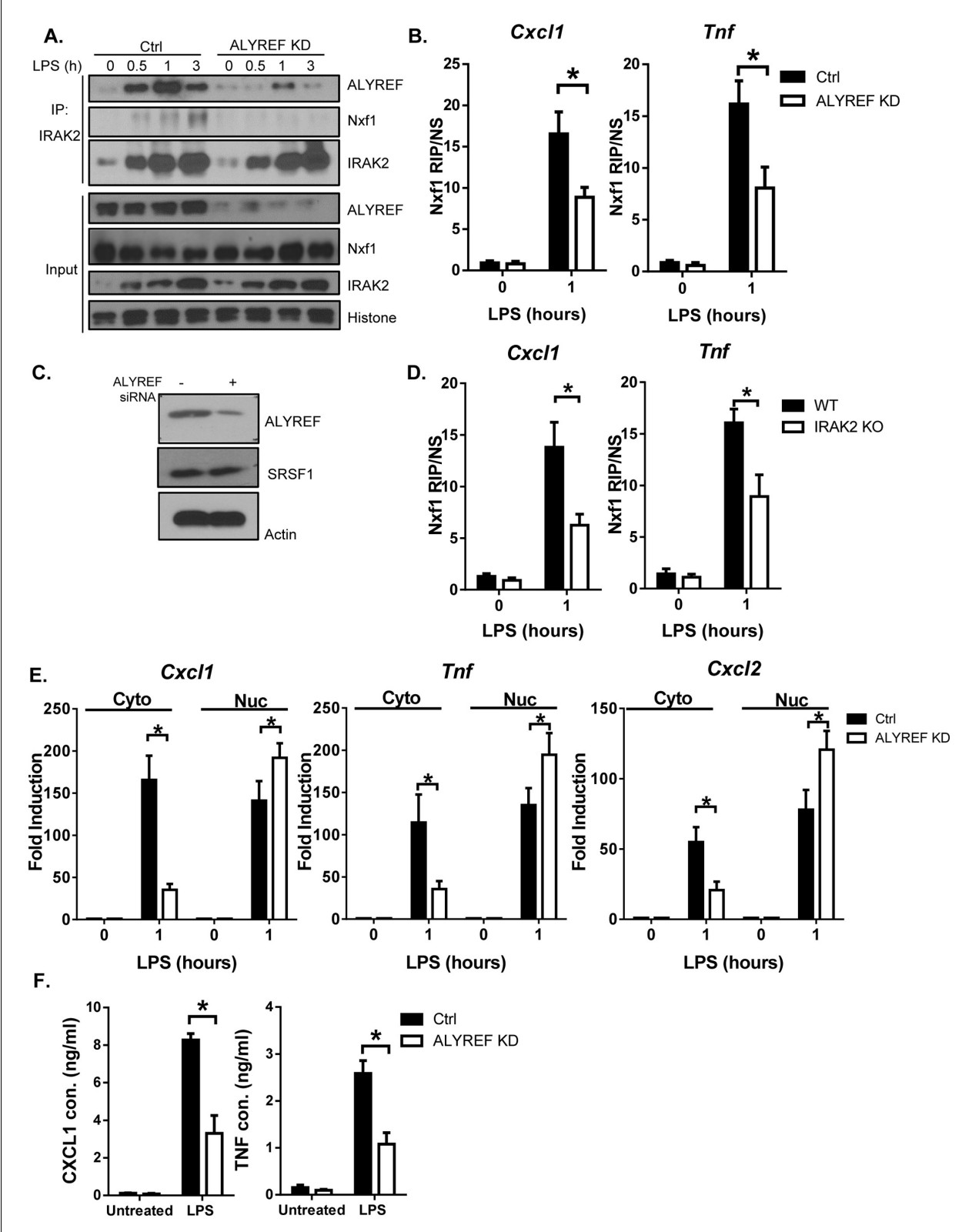

**Figure 5.** IRAK2-ALYREF facilitates the assembly of nuclear export complex on target mRNAs. (**A**) Nuclear extracts were isolated from Control (non-targeting siRNA) and ALYREF knockdown (KD, with ALYREF siRNA) macrophages treated with LPS for the indicated times, followed by immunoprecipitation (IP) with anti-IRAK2 and analyzed by western blot analysis with the indicated antibodies. (**B**) Control (non-targeting siRNA) and ALYREF knockdown (KD, with ALYREF siRNA) macrophages were treated with LPS for the indicated times, followed by RNA immunoprecipitation with

*Figure 5 continued on next page*

*Figure 5 continued*
anti-Nxf1 antibody and RT-PCR analyses of the indicated mRNAs. The presented are the relative values normalized against IgG control (Materials and methods). (C) Western analysis of lysates from cells used in *Figure 5B* with the indicated antibodies. (D) WT and IRAK2 KO BMDMs were treated with LPS for the indicated times, followed by RNA immunoprecipitation with anti-Nxf1 antibody and RT-PCR analyses of the indicated mRNAs. The presented are the relative values normalized against IgG control (Materials and methods). (E) Total mRNAs were isolated from the cytoplasmic and nuclear fractions of Control (non-targeting siRNA) and ALYREF knockdown (KD, with ALYREF siRNA) macrophages treated with LPS for indicated times were subjected to RT-PCR analyses. (F) Control (non-targeting siRNA) and ALYREF knockdown (KD, with ALYREF siRNA) macrophages were treated with LPS for 6 hr, followed by ELISA for CXCL1 and TNF levels in the supernatant. The experiments were repeated for three times. Data represent mean ± SEM; *p<0.05 by Student's t test.
DOI: https://doi.org/10.7554/eLife.29630.030
The following source data is available for figure 5:

**Source data 1.** The numerical data for the graphs in *Figure 5*.
DOI: https://doi.org/10.7554/eLife.29630.031

support of this, LPS stimulation induced Nxf1's binding to the mRNAs in an ALYREF- and IRAK2-dependent manner (*Figure 5B–D*).

To further assess the importance of ALYREF in LPS-induced IRAK2-dependent mRNA nuclear export, we measured nuclear and cytoplasmic mRNA levels in ALYREF knockdown cells. Consistent with the RIP analysis, we found that LPS-induced mRNAs of *Cxcl1*, *Cxcl2* and *Tnf* accumulated in the nucleus in the absence of ALYREF (*Figure 5E*), indicating the importance of this nuclear export factor in IRAK2-mediated nuclear export of the inflammatory mRNAs. By enzyme-linked immunosorbent assay (ELISA) analysis, we showed that LPS-induced secretion of CXCL1 and TNF was dramatically reduced in ALYREF knockdown cells (*Figure 5F*).

## The SRSF1-binding site in the target mRNAs allows IRAK2-dependent nuclear export

Since the LPS-induced mRNAs that are dependent on IRAK2 for nuclear export were enriched with SRSF1-binding motif in the 3'UTRs (*Figure 3C*), we decided to directly test the importance of the SRSF1-binding sites in the target mRNAs for their nuclear export. Using *Cxcl1* 3'UTR (nt 781–901, which contains stimulus-sensitive motifs (*Datta et al., 2010*; *Hartupee et al., 2007*), we validated the SRSF1 binding to this transcript by RNA electrophoretic mobility shift assay (REMSA) (*Figure 6A*). We identified several putative SRSF1-binding motifs in this region of *Cxcl1* 3'UTR. Deletion analysis and REMSA showed the importance of nt 800–855 in Cxcl1 3'UTR for SRSF1 binding. We generated luciferase reporter constructs by cloning the 3'UTR of *Cxcl1* with or without deletion of the SRSF1-binding region downstream of Luciferase reporter (under the control of CMV promoter). These reporter constructs were introduced into IRAK2-deficient cells (IRAK deficient 293-IL1R cells; [*Yao et al., 2007*]) with or without co-transfection with IRAK2. While SRSF1 RIP precipitated the luciferase reporter mRNA containing the 3'UTR of *Cxcl1* (nt 721–940), IRAK2 expression reduced SRSF1 binding to the target mRNA (*Figure 6B*). However, SRSF1 RIP failed to precipitate the luciferase mRNAs with the mutated 3'UTR of *Cxcl1* (Δ790–840 and Δ829–835) (*Figure 6B*).

We then assessed the SRSF1 site-mutant constructs by luciferase reporter assay. For the cells transfected with wild-type *Cxcl1* luciferase construct (with wild-type 3' UTR of *Cxcl1* 721–940), IRAK2 strongly promoted the luciferase activity (*Figure 6C*). Consistently, the cytosolic luciferase reporter mRNAs (with wild-type 3' UTR of *Cxcl1* 721–940) were increased in the IRAK2-transfected cells compared to the cells without IRAK2 (*Figure 6D*). IRAK2 had much reduced ability to promote luciferase activity in cells transfected with mutant *Cxcl1* luciferase constructs (*Cxcl1Δ790–840* and *Cxcl1Δ829–835*) (*Figure 6C*). Removal of the functional SRSF1-binding site trapped the luciferase reporter mRNAs (*Cxcl1Δ790–840* and *Cxcl1Δ829–835*) in the nucleus and IRAK2 can no longer promote nuclear export of these mutant reporter mRNAs (*Cxcl1Δ790–840* and *Cxcl1Δ829–835*) (*Figure 6D*). These results suggest that while SRSF1 binding may sequester the luciferase reporter mRNAs in the nucleus, the SRSF1-binding sequence is actually required for nuclear export promoted by IRAK2.

Similar results were obtained in wild-type and IRAK2-deficient macrophages. LPS stimulation induced luciferase activity from the cells transfected with constructs containing wild-type 3'UTR of *Cxcl1* (nt 721–940), which was abolished in IRAK2-deficient macrophages (*Figure 6E*). However, LPS-induced luciferase activity was substantially reduced in wild-type, IRAK2-deficient or IRAK2-

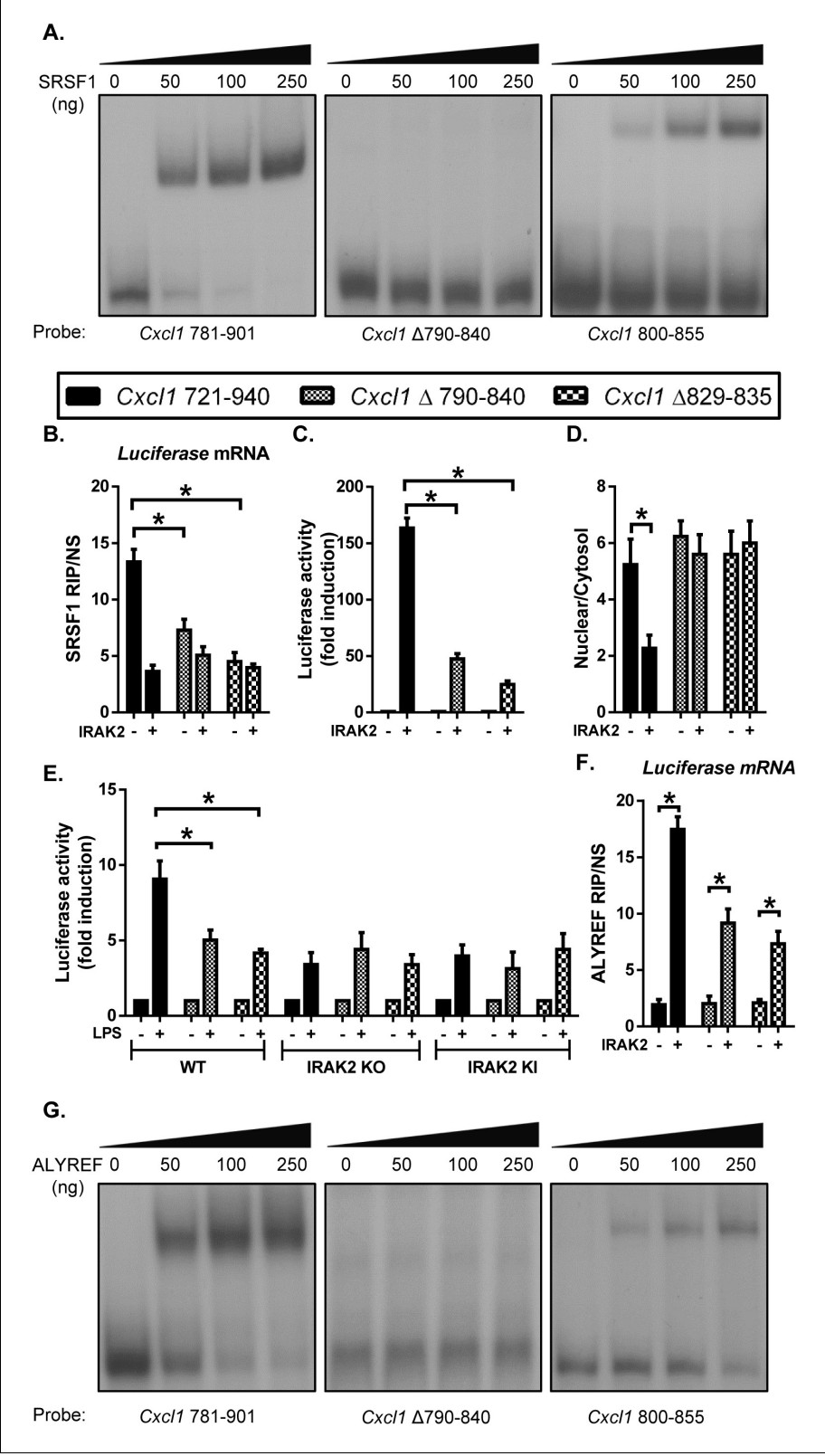

**Figure 6.** SRSF1 binding renders the LPS-sensitivity of the target mRNAs for their nuclear export. (**A**) REMSA was performed to show binding of purified recombinant SRSF1 protein to the *Cxcl1* 3'UTR (nt 781–901), *Cxcl1* 3'UTR (nt Δ 790–840 in 781–901) and *Cxcl1* 3'UTR (800-855). (**B–D**) CMV-Luc- *Cxcl1* 3'UTR (721-940), (nt Δ 790–840 in 721–940) and (nt Δ 829–835 in 721–940) plasmids were transiently transfected in IRAK-deficient 293-IL1R (I1A) cells for
*Figure 6 continued on next page*

*Figure 6 continued*

24 hr; followed by RNA immunoprecipitation with anti-SRSF1 antibody and RT-PCR analyses of the indicated mRNAs, the presented are the relative values normalized against IgG control (Material and methods) (B); Luciferase activity was measured (C); RT-PCR analysis of total mRNAs isolated from the cytoplasmic and nuclear fractions (D). (E) CMV-Luc- *Cxcl1* 3'UTR (721-940), (nt Δ 790–840 in 721–940) and (nt Δ 829–835 in 721–940) plasmid were transiently transfected in wild-type, IRAK2 KO and IRAK2 KI macrophages and treated with LPS for 12 hr, followed by luciferase assay. (F) For the transfected cells as described in *Figure 6B–D*, cell lysates were subjected to RNA immunoprecipitation with anti-ALYREF antibody and RT-PCR analyses of the indicated mRNAs. The presented are the relative values normalized against IgG control (Materials and methods). (G) REMSA was performed to show binding of purified recombinant ALYREF protein to the *Cxcl1* 3'UTR (nt 781–901), *Cxcl1* 3'UTR (nt Δ 790–840 in 781–901) and *Cxcl1* 3'UTR (800-855). Data represent mean ± SEM; *p<0.05 by Student's t test.

DOI: https://doi.org/10.7554/eLife.29630.032

The following source data and figure supplement are available for figure 6:

**Source data 1.** The numerical data for the graphs in *Figure 6*.

DOI: https://doi.org/10.7554/eLife.29630.034

**Figure supplement 1.** Coomassie blue staining of purified recombinant ALYREF and SRSF1.

DOI: https://doi.org/10.7554/eLife.29630.033

kinase inactive macrophages transfected with mutant luciferase constructs (Δ790–840 and Δ829–835) (*Figure 6E*). These results confirm that IRAK2-mediated nuclear export of the target mRNAs is dependent on the SRSF1-binding sites in the target mRNA.

Consistent with the fact that IRAK2 is required for LPS-induced ALYREF's binding to the target mRNAs, IRAK2 promoted ALYREF RIP of wild-type luciferase reporter mRNAs (containing wild-type 3'UTR of *Cxcl1* nt 721–940) (*Figure 6F*). Interestingly, ALYREF reduced the binding to the mutant luciferase reporter mRNAs (Δ790–840 and Δ829–835) even in the presence of IRAK2 (*Figure 6F*). These results suggest that the SRSF1-binding site seems to be required for ALYREF's binding to the target mRNAs. In support of this, ALYREF can directly bind to the probe that contains the SRSF1-binding site in the in vitro RNA-binding assay (*Cxcl1* nt 800–855, *Figure 6G* and *Figure 6—figure supplement 1*). Taken together, our results implicate that the SRSF1-binding site renders the LPS-IRAK2 sensitivity of the target mRNA for the nuclear export possibly via the IRAK2-mediated removal of SRSF1 and recruitment of nuclear export factors ALYREF to the target mRNAs.

## IRAK2 kinase-inactive knock-in mice are resistant to LPS-induced septic shock

Mirroring the IRAK2 knockout cells, the LPS-induced inflammatory mRNAs were accumulated in the nucleus in IRAK2 kinase-inactive knockin macrophages compared to that in wild-type cells (*Figure 3B*). Consistent with the defect in promoting nuclear export of target mRNAs, LPS-induced ALYREF RIP of inflammatory mRNAs was abolished in IRAK2 kinase-inactive knockin macrophages (*Figure 4F*). Finally, ELISA analysis validated the importance of IRAK2 kinase activity for the production of LPS-induced CXCL1, CXCL2 and TNF (*Figure 1D*). Since IRAK2 kinase-inactive knock-in macrophages mirrored IRAK2 knockout cells in all the ex vivo experiments in response to LPS stimulation, we then further investigated the impact of this IRAK2 mutation on LPS-response in vivo. We found that IRAK2 kinase-inactive knockin mice showed substantially increased survival after LPS-induced septic shock compared to that of the wild-type mice (*Figure 7A*)(*Mao et al., 2013*). Likewise, LPS-induced serum CXCL1 and TNF levels were much reduced in IRAK2 kinase-inactive knockin mice compared to that of the wild-type mice (*Figure 7B*).

## Discussion

While TLR-induced inflammatory gene expression is essential for host defense against infections of pathogens, dysregulated production of cytokines and chemokines is detrimental to the host, resulting in septic shock and other inflammatory diseases. Nuclear and cytoplasmic compartments enable spatial separation and regulation of transcription and translation. In this study, we report that LPS/TLR4 engagement activates a nuclear function of IRAK2 that facilitates the assembly of nuclear export machinery to enable export the inflammatory mRNAs to the cytosol for protein translation.

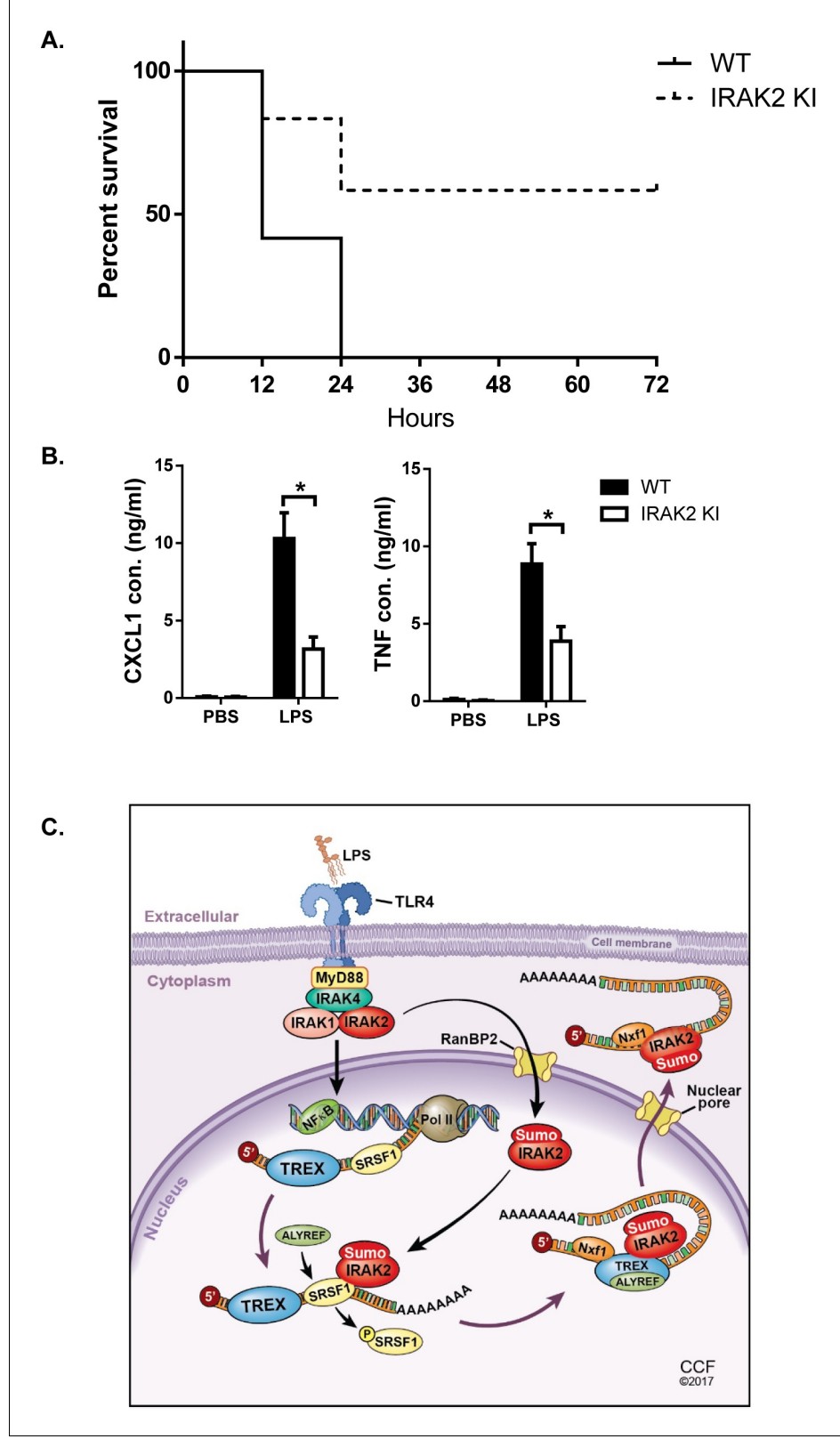

**Figure 7.** IRAK2 kinase-inactive knock-in mice are resistant to LPS-induced septic shock. 10 pairs Wild-type (WT) and IRAK2 kinase-inactive (KI) mice were injected intraperitoneally with 10 mg LPS/kg, and the survival of injected mice was monitored for 72 hr. (A) The survival curves were created by the method of Kaplan and Meier. (B) ELISA was performed for the levels of TNF and CXCL1 in the plasma 4 hr after LPS challenge. Data represent

*Figure 7 continued on next page*

*Figure 7 continued*

mean ± SEM; *p<0.05 by Student's t test. (**C**) Model for the nuclear function of IRAK2. Upon LPS stimulation, IRAK2 is activated, followed by RanBP2-mediated sumoylation and subsequent nuclear translocation. Nuclear IRAK2 then phosphorylates SRSF1 and reduces SRSF1 binding to the target mRNAs which promotes the RNA binding of the nuclear export adaptor ALYREF and export receptor Nxf1 loading for export of mRNAs.

DOI: https://doi.org/10.7554/eLife.29630.035

The following source data is available for figure 7:

**Source data 1.** The numerical data for the graphs in **Figure 7**.

DOI: https://doi.org/10.7554/eLife.29630.036

IRAK2 kinase activity is required for RanBP2-mediated IRAK2 sumoylation and nuclear translocation. Array analysis identified an SRSF1-binding motif enriched in the mRNAs that are accumulated in the nucleus of IRAK2-deficient and IRAK2-kinase inactive macrophages. Nuclear IRAK2 then phosphorylates SRSF1 and reduces SRSF1 binding to the target mRNAs. At the same time, LPS induces interaction between IRAK2 and nuclear export adaptor ALYREF to replace SRSF1 on the target mRNAs. The recruited ALYREF in turn bridges the interaction between the mRNA targets and nuclear export receptor Nxf1, triggering nuclear export receptor Nxf1 loading for the export of the mRNAs. (**Figure 7C**) Therefore, LPS-induced IRAK2 activation provides a critical check point for the production of inflammatory cytokines and chemokines by controlling the nuclear and cytoplasmic distribution of the LPS-induced transcripts.

IRAK2 is known as an atypical kinase due to the amino acid substitution of key catalytic residues (Asp -> Asn[333]; Asp ->His[351]). Notably, atypical kinases such as KSR2 (**Brennan et al., 2011**) and CASK (**Mukherjee et al., 2008**) have been reported to carry catalytic activity. We found recombinant IRAK2 was able to autophophorylate and phosphorylate MBP and SRSF1; mutations in the ATP-binding site and catalytic residues impaired the function of IRAK2. IRAK2 kinase-inactive knockin mice showed substantially increased survival after LPS-induced septic shock compared to that of the wild-type mice. Likewise, LPS-induced serum CXCL1 and TNF levels were much reduced in IRAK2 kinase-inactive knockin mice compared to that of the wild-type mice. Taken together, these results indicate that the IRAK2 kinase activity is critical for the function of IRAK2 in LPS-induced pro-inflammatory response.

While LPS-induced IRAK2 modification was previously reported, we now identified RanBP2 as the E3 ligase for IRAK2 sumoylation. We also for the first time report that LPS induces IRAK2 nuclear translocation, which seems to be dependent on LPS-induced IRAK2 sumoylation. IRAK2 mutants defective in sumoylation were retained in the cytoplasm. Likewise, LPS-induced IRAK2 nuclear translocation was abolished in RanBP2 knock-down cells, supporting the critical role of IRAK2 sumoylation for its nuclear translocation. Consistently, RanBP2 was shown to modulate cytoplasmic and nuclear transport of macromolecules. RanBP2 is a major nucleoporin that extends cytoplasmic filaments from the nuclear pore complex and contains phenylalanine–glycine repeats that bind the transport receptor importin-β. Indeed, IRAK2 was also shown to interact with importin-β, which requires the NLS on IRAK2. It is intriguing that although IRAK2 kinase inactive mutants still interact with importin-β, they were unable to interact with RanBP2 and failed to translocate into the nucleus. While these results suggest a potential impact of IRAK2 kinase activity on RanBP2, IRAK2 autophosphorylation may affect its interaction with RanBP2. Our data showed that mutations in putative IRAK2 phosphorylation sites S136/T140 (**Weintz et al., 2010**) substantially reduced the interaction of IRAK2 with RanBP2. It suggests that the activation of IRAK2 may result in auto-phosphorylation at S136A and T140A, which in turn mediates the interaction with RanBP2. Future studies are required to elucidate the detailed mechanism for how phosphorylation of IRAK2 affects its interaction with RanBP2 and subsequent sumoylation.

In addressing the functional impact of LPS-induced IRAK2 nuclear translocation, we found that IRAK2 plays a critical role in promoting nuclear export of LPS-induced transcripts. By array analysis, a SRSF1-binding motif was found to be enriched in the mRNAs targets that are dependent on IRAK2 for nuclear export. As discussed above, recombinant IRAK2 was able to phosphorylate SRSF1 and LPS-activated IRAK2 can phosphorylate SRSF1 and thereby reduces SRSF1 binding to the target mRNAs in macrophages. These results suggest that SRSF1 binding endows target mRNAs with

sensitivity for LPS to promote nuclear export, and that LPS induces nuclear function of IRAK2 to mediate the removal of SRSF1 from the target mRNAs. Importantly, Mass Spec analysis showed that IRAK2 also interacts with nuclear export adaptor ALYREF in addition to importin-β and RanBP2. Notably, ALYREF and SRSF1 actually belong to the same family of proteins called the shuttling SR (serine- and arginine-rich) proteins, sharing a similar RNA-binding domain. Indeed, we found SRSF1 and ALYREF were able to bind to the same SRSF1-binding motif; and the removal of SRSF1 allowed nuclear export adaptor ALYREF binding to the target mRNAs, suggesting that SRSF1-mediated nuclear sequestration of target mRNAs might be achieved by blocking the binding of ALYREF to the mRNAs. Importantly, previous studies have reported that ALYREF (a subunit of the so-called TREX complex for nuclear export) makes contact with nuclear export receptor Nxf1, serving as a bridge between mRNA and nuclear export receptor Nxf1. Thus, nuclear IRAK2 mediates the removal of SRSF1 and facilitates the assembly of nuclear export machinery to export the inflammatory mRNAs to the cytosol for translation. Interestingly, about 10–30% LPS-induced inflammatory transcripts were exported out of the nucleus in the absence of IRAK2 (*Figure 3B*), which was consistent with the residual ALYREF's binding to these transcripts in IRAK2-/- cells (*Figure 4D*). These results implicate that there might be yet another mechanism besides IRAK2 that allows ALYREF binding to the target mRNAs for nuclear export. Future studies are required to identify additional players that modulate TLR-induced nuclear/cytosol distribution of inflammatory transcripts.

## Materials and methods

### Animals

IRAK2-deficient mice were previously described (*Wan et al., 2009*). IRAK2 kinase-inactive knockin (K235A and K236A) mice were generated by co-microinjection of in vitro-translated Cas9 mRNA and gRNA into the C57BL/6 zygotes. The gRNA sequence used to generate the knockin mice is GA TCACTAATACGACTCACTATAGGCCTCCCTGAGCTTCTTGAGTTTTAGAGCTAGAAAT and GA TCACTAATACGACTCACTATAGGTTCGCCTTCAAGAAGCTCGTTTTAGAGCTAGAAAT. All procedures using animals were approved by the Cleveland Clinic Institutional Animal Care and Use Committee (Protocol Number: 2014–1229 and 2017–1814).

### Biological reagents and cell culture

LPS (Escherichia coli 055:B5) was purchased from Sigma-Aldrich (St. Louis, MO). Oxidized low density lipoprotein (ox-LDL) was purchased from ThermoFisher (Waltham, MA). R848 and CpGB were purchased from Invivogen. Anti-IRAK2 (ab62419, RRID:AB_956084) and ant-Importin-β (ab2811, RRID:AB_2133989) antibodies were purchased from Abcam (United Kindom). Antibodies to RanBP2 (sc-74518, RRID:AB_2176784), ALYREF (sc-32311, RRID:AB_626667), Actin (sc-1615, RRID:AB_ 630835) and Tubulin (sc-8035, RRID:AB_628408) were purchased from Santa Cruz (Dallas, TX). Antibodies to SRSF1 (14902), NXF1 (12735), Histone (4499) and SUMO1 (4930) were purchased from Cell Signaling (Danvers, MA). SRSF1 recombinant protein (ab219488) was purchased from Abcam. Myelin Basic Protein (MBP) (31314) was purchased from Active Motif (Carlsbad, CA). Bone-marrow derived macrophages were obtained from the bone marrow of tibia and femur by flushing with DMEM. The cells were cultured in DMEM supplemented with 20% fetal bovine serum (FBS), penicillin G (100 μg/ml), streptomycin (100 μg/ml) with M-CSF (50 ng/ml) for seven days before the experiments. Hela cells (RRID: CVCL_0030) were purchased from ATCC (Manassas, VA) and authenticated by STR analysis. The Heal cells were cultured in DMEM supplemented with 10% FBS, penicillin G (100 μg/ml) and streptomycin (100 μg/ml) and negative for mycoplasma contamination test using a PCR detection method.

### Protein purification

His-tagged IRAK2 WT, K123/182/592R, KK235AA, H351A mutant and Ally were sub-cloned into PET28a vector and expressed in *E. Coli* (BL21). The recombinant protein was purified using Ni Sepharose 6 Fast Flow column (GE HealthCare). Size exclusion chromatography on a Superdex s200 high resolution column (GE HealthCare, United Kingdom) was used as a final step for purification. Recombinant Aos1/Uba2, UBC9, RanBP2$_{RB3-4}$ and SUMO1 were purified as previously described (*Werner et al., 2012*).

## In vitro kinase assay

IRAK2 WT, KK235AA and H351A (100 nM) was incubated with myeline basic protein (MBP) (10 nM) in the kinase assay buffer containing 25 mM Tris (pH 7.5), 5 mM β-glycerophosphate, 2 mM DTT, 0.1 mM $Na_3VO_4$, 10 mM $MgCl_2$ supplemented with 100 nM ATP and 1 ul [γ-$^{32}$P]-ATP (PerkinElmer) (10 μCi) at 37°C for 30 min. The samples were subjected to SDS-PAGE followed by autoradiograph.

## In vitro sumoylation assay

IRAK2 WT and K123/182/592R (200 nM) was incubated with 50 nM Aos1/Uba2, 100 nM Ubc9, 24 nM RanBP2$_{RB3-4}$, 9 μM SUMO1, and 1 mM ATP in a modified sumoylation assay buffer containing 20 mM PIPES (pH 6.8), 150 mM NaCl, 1 mg/ml ovalbumin, 0.05% Tween 20, 1 mM DTT, and 1 μg/ml of each aprotinin, leupeptin, and pepstatin at 30°C. Samples were analyzed by western blot. RanBP2 level was determined prior to the reaction.

## In vitro desumoylation assay

SUMO protease Ulp1 (12588018) was purchased from ThermoFisher. Sumolyated-IRAK2 was incubated with SUMO protease in the buffer containing 50 mM Tris-HCl, pH 8.0, 0.2% NP-40, 150 mM NaCl, 1 mM DTT at 30°C for 30 min.

## Immunoblot, immunoprecipitation and nuclear fractionation

Cells were harvested and lysed on ice in a lysis buffer containing 0.5% Triton X-100, 20 mM Hepes pH 7.4, 150 mM NaCl, 12.5 mM -glycerophosphate, 1.5 mM MgCl2, 10 mM NaF, 2 mM dithiothreitol, 1 mM sodium orthovanadate, 2 mM EGTA, 20 mM aprotinin, and 1 mM phenylmethylsulfonyl fluoride for 20 min, followed by centrifuging at 12,000 rpm for 15 min to extract clear lysates. For immunoprecipitation, cell lysates were incubated with 1 μg of antibody and A-sepharose beads at 4 degree overnight. After incubation, the beads were washed four times with lysis buffer and the precipitates were eluted with 2x sample buffer. Elutes and whole cell extracts were resolved on SDS-PAGE followed by immunoblotting with antibodies. Nuclear fractionation was performed using NUCLEI EZ PREP kit purchased from Sigma-Aldrich in accordance with the manufacturer's instruction. Nuclear pellets were suspended in 30 μl of nuclear extraction buffer (20 mM HEPES, 400 mM NaCl, 1 mM EDTA, 1 mM EGTA in water, pH 7.9) containing freshly prepared 1 mM DTT and protease inhibitor cocktail. After 1.5 hr incubation on ice bath with intermittent vortexing, extracts were centrifuged and supernatant was collected for immunoprecipitation.

## SiRNA-mediated knockdown

siGENOME SMARTpool siRNAs were purchased from Dharmacon (Lafayette. CO) for RanBP2, ALYREF and SRSF1 knockdown in macrophages. The targeted sequences are listed below: RanBP2, GCACAUGUUGUUAAACUUA, GAGACGAGAGCAAGUAUUA, GAAUUAAACCCAACGCAAA, GAGCUUUACCGUUCAAAUA; ALYREF, CGAAACAACUUCCCGACAA, UCAUUAAGCUGAACCGGAG, UGAAUUUGGGACAUUGAAA, AGACCUGCACAGAGCAUAA; SRSF1, GAAAGAAGAUAUGACGUAU, GCACUGGUGUCGUGGAGUU, UAUGUUACGCUGAUGUUUA, GAAGCUGGCAGGACUUAAA. siGENOME Non-targeting siRNA Pools were used for the control groups. Amaxa Cell Line Nucleofector Kit V (LONZA, Switzerland) was used to transfect macrophages following manufacturer's instructions.

## ELISA assay

Supernatants from cell cultures were collected and measured for the level of mouse cytokines CXCL1 and TNF using Duoset ELISA kits (R&D system, Minneapolis, MN) according to manufacturer's instructions.

## RNA immunoprecipitation (RIP)

The RIP assay was performed following the protocols as previous described (Herjan et al., 2013; Keene et al., 2006). Briefly, 10$^7$ macrophages were left untreated or treated with LPS (1 μg/ml) for 1 hr. Cells were washed three times with ice cold PBS and suspended in the lysis buffer. The lysate was centrifuged and the supernatant was immunoprecipitated overnight at 4 Celsius degree, using Dynabeads (Invitrogen) preincubated with 20 μg anti-IRAK2, SRSF1, ALYREF, Nxf1 antibodies or

anti-IgG antibody. RNA was purified from immunoprecipitates with Trizol (Invitrogen) according to the manufacturer's instructions and treated with RNase-free DNase, the cDNAs were synthesized and 10% of the reverse transcriptase product was subjected to quantitative real-time PCR. RIP data analysis: Ct value of each RIP RNA fraction was normalized to the Input RNA fraction Ct value for the same qPCR Assay (ΔCt) to account for RNA sample preparation differences. Then the normalized RIP fraction Ct value (ΔCt) was adjusted for the normalized background (anti-IgG) [non-specific (NS) Ab] fraction Ct value (ΔΔCt). The fold enrichment [RIP/non-specific (NS)] was calculated by linear conversion of the ΔΔCt. Below are the formulas used for the calculation: ΔCt [normalized RIP]=Ct [RIP] – (Ct [Input] – Log2 (fraction of the input RNA saved))); ΔΔCt [RIP/NS] = ΔCt [normalized RIP] – ΔCt [normalized NS]; Fold Enrichment = 2 (-ΔΔCt [RIP/NS]).

## RNA Electrophoretic Mobility Shift Assay (REMSA)

Increasing amounts of purified protein and labeled probes (10 fmol, see in vitro transcription) were combined in the binding buffer for 30 min. The final REMSA-binding buffer concentrations were 140 mM KCl, 10 mM HEPES pH 7.9, 5% glycerol, 1 mM DTT and 0.33 mg/ml tRNA. The reaction was further supplemented with 15 µg salmon sperm DNA to reduce non-specific interactions from the lysate. Complexes were resolved on either 4% or 6% non-denaturing polyacrylamide gels.

## In vitro transcription

REMSA-radiolabeled 3' UTR RNA probes were synthesized from BamHI linearized plasmids templates with T7 RNA polymerase using 1 mM GTP, 1 mM ATP, 1 mM CTP, 0.005 mM UTP and 25 µCi of $^{32}$P-labeled UTP for 3 hr at 37°C. Probes were DNAase I treated for 20 min and then phenol:chloroform extracted. The aqueous phase was passed through a Micro Bio-Spin P30 column according to manufacturer's instructions (BioRad).

## Proximity Ligation Assay (PLA)

PLA was performed using the Duolink In Situ Green Kit purchased from Sigma-Aldrich (DUO92101) in accordance with the manufacturer's instruction. Briefly, transfected cells were washed once with ice cold PBS, followed by fixation with 4% paraformaldehyde for 15 min at room temperature. Fixed cells were then washed three times with PBS and permeabilized with 0.3% Triton X-100 containing PBS for 10 min. Permeabilized cells were blocked with 5% normal goat serum for 1 hr at room temperature. The cells were then incubated with primary antibodies diluted in 10% normal goat serum supplemented with 0.1% Tween at 4°C overnight. Following the incubation, the cells were washed three times with PBS and then incubated with two PLA probes (Duolink In Situ PLA Probes Anti-rabbit PLUS and Anti-Mouse MINUS, Sigma-Aldrich) for 1 hr at 37°C. After probe incubation, the samples were incubated in ligation solution for 1 hr at 37°C. After ligation, cells were washed with Wash Buffer A and incubated in the amplification solution for 2 hr at 37°C. Cells were then serially washed twice in 1 × Wash Buffer B, 0.01 × Wash Buffer B once, and PBS once, followed by incubation with secondary antibodies for 1 hr at room temperature. Finally, cells were washed three times with PBS and mounted in Duolink In Situ Mounting Medium supplemented with DAPI. Fluorescence images were obtained with a confocal microscope.

## Real-time PCR

Total RNA was extracted from spinal cord with TRIzol (Invitrogen, Carlsbad, CA) according to the manufacturer's instructions. 1 µg total RNA for each sample was reverse-transcribed using the SuperScript II Reverse Transcriptase from Thermo Fisher Scientific. The resulting complementary DNA was analyzed by real-time PCR using SYBR Green Real-Time PCR Master Mix. All gene expression results are expressed as arbitrary units relative to expression Actin.

## Microarray analysis

Cytoplasmic and nuclear RNA were extracted from WT and IRAK2-deficient macrophages and subjected to microarray analysis. Targets preparation was performed on a Biomek FXP (Beckman Coulter) using a GeneChip HT 3'IVT Express Kit (Affymetrix, Santa Clara, CA) in accordance with the manufacturer's instruction. Labeled cRNA were hybridized on an Affymetrix GeneChip HT-MG-430PM-96 (Affymetrix). Array hybridization, washing, and scanning were performed on GeneTitan

(Affymetrix). Three independent biological replicates were analyzed in each experiment. Probe signals were subjected to presence calling and normalized to derive relative transcript abundance. The ratio (R) between the nuclear (N) and cytoplasmic (C) abundance of each transcript is calculated (R = N/C) to quantify the nuclear retention of the transcript. The nuclear/cytoplasmic ratio of the transcript in the LPS-treated IRAK2 knockout BMDMs (RKO) is then divided by that of the same transcript in the corresponding wild-type BMDMs (RWT). The resulting value is used as an index (I = RKO/ RWT) to measure the impact of IRAK2 deficiency on the nuclear retention of the transcript. Transcripts induced by LPS treatment are ranked based on the index I and the top 70 transcripts are used as the positive set for motif enrichment analysis.

## Motif enrichment analysis

To define the sequences, mouse (Mus Musculus) genes were downloaded from Ensembl BioMART in June 2014 and represent the GRCm38.p2 release of gene models. When there are multiple isoforms for the same gene we used the longest isoform to define its mature mRNA sequence and the genomic locus covered by its pre-mRNA sequence. We performed RNA-READ algorithm (unpublished work, XL, HDL, and QM, in preparation) to perform motif enrichment test. Specifically, we performed a likelihood ratio test to assess whether any of the previously defined RBP motifs (*Ray et al., 2013*) from our collection could better distinguish the positive set from the negative set when provided to a regression algorithm that also had access to a control set of features that consisted of all the dinucleotides contained within the corresponding motif as well as the length of the target sequence; the construction of these regression models is described below. The comparisons between the motif and the control features were restricted to specific regulatory region of the transcripts (i.e., 3' UTR, 5'UTR or the coding region). We scored each regulatory region using a given motif by summing the accessibility of all the target sites, where a target site was defined as a perfect match to the IUPAC representation of the motif and the accessibility of a target site was defined as the average single base accessibility of the bases in the site. A score of zero was assigned to those regulatory region did not contain a motif match. The single base accessibility was assessed using RNAplfold (*Bernhart et al., 2006*) as described previously (*Li et al., 2010*). We used the parameters with W = 200, L = 150 and U = 1. Although the analysis was applied in specific regulatory region, the entire transcript was input into RNAplfold to ensure correct folding of the bases close to the start codon and stop codon. We used the glmnet.R package (*Friedman et al., 2010*) to apply Lasso penalized logistic regression. In the Lasso regression, the hyper-parameter lambda (i.e. the regularization strength) was selected through a five-fold cross-validation procedure, from the lambda sequence computed by glmnet using the default settings of nlambda and lambda.min.ratio. The final value for lambda was the one (from the sequence) with the smallest average generalization error across the five folds. We then used this value of lambda with the 'glmnet.fit' object on the entire dataset to compute the weights for the features. The features with non-zero weights were selected as contributing most to the prediction. After the non-zero weight features were defined, we trained two standard logistic regression models: one using all non-zero weight features (including the motif) and one that contained only the non-zero weighted control features, and then assessed whether there was a significant difference in predictive power between these two nested models using a log-likelihood ratio test.

## LPS-induced septic shock

Ten pairs of Wildtype and IRAK2 KI mice were injected intraperitoneal with 10 mg of LPS/kg, and survival was monitored for 72 hr. Blood for determination of plasma TNF and CXCL1 was obtained from the tail vein 4 hr after challenge.

## Statistical analysis

The significance of differences between two groups was determined by Student's t-test (two-tailed). A P value less than 0.05 was considered significant. The survival curves were created by the method of Kaplan and Meier. For statistical comparison, survival curves were analyzed using the log rank test.

## Acknowledgement

This work is supported by grants from NIH (2PO1HL029582 and PO1CA062220) and National MS society (RG5130A2/1). HZ is supported by Postdoctoral Research Fellow Award (1–16-PDF-138) from American Diabetes Association. KB is supported by the National Science Centre, Poland (2015/19/B/NZ6/01578).

## Additional information

### Funding

| Funder | Grant reference number | Author |
| --- | --- | --- |
| National Multiple Sclerosis Society | RG5130A2/1 | Xiaoxia Li |
| National Institutes of Health | 2PO1HL029582 | Xiaoxia Li |
| American Diabetes Association | Postdoctoral Research Fellow Award,1-16-PDF-138 | Hao Zhou |
| Narodowe Centrum Nauki | 2015/19/B/NZ6/01578 | Katarzyna Bulek |
| National Institutes of Health | PO1CA062220 | Xiaoxia Li |

The funders had no role in study design, data collection and interpretation, or the decision to submit the work for publication.

### Author contributions

Hao Zhou, Conceptualization, Data curation, Formal analysis, Funding acquisition, Validation, Investigation, Visualization, Methodology, Writing—original draft, Writing—review and editing; Katarzyna Bulek, Tomasz Herjan, Conceptualization, Data curation, Formal analysis, Validation, Investigation, Visualization, Methodology, Writing—original draft, Writing—review and editing; Xiao Li, Conceptualization, Software, Formal analysis, Validation, Investigation, Visualization, Methodology, Writing—original draft, Writing—review and editing; Minjia Yu, Wen Qian, Data curation, Formal analysis, Validation, Investigation; Han Wang, Xing Chen, Formal analysis, Investigation; Gao Zhou, Software, Investigation, Visualization; Hui Yang, Formal analysis, Investigation, Visualization; Lingzi Hong, Data curation, Formal analysis; Junjie Zhao, Formal analysis, Investigation, Visualization, Writing—review and editing; Luke Qin, Data curation; Koichi Fukuda, Conceptualization, Investigation; Annette Flotho, Resources, Investigation; Ji Gao, Ashok Dongre, Julie A Carman, Data curation, Investigation; Zizhen Kang, Bing Su, Frauke Melchior, Resources, Supervision; Timothy S Kern, Jonathan D Smith, Supervision; Thomas A Hamilton, Paul L Fox, Conceptualization, Supervision, Writing—review and editing; Xiaoxia Li, Conceptualization, Data curation, Formal analysis, Funding acquisition, Investigation, Methodology, Writing—original draft, Project administration, Writing—review and editing

### Author ORCIDs

Hao Zhou http://orcid.org/0000-0001-9109-2489
Frauke Melchior http://orcid.org/0000-0001-9546-8797
Xiaoxia Li http://orcid.org/0000-0002-4872-9525

### Ethics

Animal experimentation: This study was performed in strict accordance with the recommendations in the Guide for the Care and Use of Laboratory Animals of the National Institutes of Health. All procedures using animals were approved by the Cleveland Clinic Institutional Animal Care and Use Committee (Protocol Number: 2014-1229 and 2017-1814).

### Decision letter and Author response

Decision letter https://doi.org/10.7554/eLife.29630.038
Author response https://doi.org/10.7554/eLife.29630.039

## Additional files

**Supplementary files**
• Transparent reporting form
DOI: https://doi.org/10.7554/eLife.29630.037

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
