## [Decision Letter]

Thank you for submitting your article "IRAK2 directs stimulus-dependent nuclear export of inflammatory mRNAs" for consideration by *eLife*. Your article has been favorably evaluated by Tadatsugu Taniguchi (Senior Editor) and three reviewers, one of whom is a member of our Board of Reviewing Editors.

The reviewers have discussed the reviews with one another and the Reviewing Editor has drafted this decision to help you prepare a revised submission.

Summary:

TLR signaling critically regulates the expression of pro-inflammatory mediators including cytokines and chemokines. Tremendous efforts have been put on investigating the underlying mechanisms at transcriptional and post-transcriptional levels. It is known that binding to chemokine/cytokine mRNAs to the SRSF1 (AF2/ASF) protein mediates decay of these mRNAs in the nucleus, but the molecular mechanism by which pro-inflammatory mRNAs are regulated during nuclear export remains unknown. It is also known that IRAK2, downstream of the TLRs, plays a critical role in regulating chemokine/cytokine expression, but the exact mechanism is not defined. In the present study, the authors performed comprehensive analyses using multiple state-of-the-art approaches and demonstrate for the first time that activation of TLR4 induces nuclear localization of IRAK2 to promote nuclear export of pro-inflammatory mRNAs (CXCL1, TNF and CXCL2) for ribosomal translation. Interestingly, IRAK2 kinase activity is essential for its nuclear translocation via LPS-induced RanBP2-mediated sumoylation of IRAK2. The nuclear IRAK2 then phosphorylates SRSF1 to reduce its binding to the target mRNAs. Meanwhile, IRAK2 also assembles a complex with the nuclear export adaptor ALYREF, which promotes the binding of those cytokine mRNAs to ALYREF and facilitates loading of the nuclear export receptor NXF1 to the mRNAs for nuclear export of those inflammatory chemokine/cytokine mRNAs. These novel and interesting findings answer a longstanding question in the field. The study is well designed and the conclusion is supported by extensive convincing biochemical and cell biological analyses. The discoveries of IRAK2 sumoylation and its regulation of nuclear export of the target chemokine/cytokine mRNAs represent a major step-forward in our understanding TLR and innate immune signal transduction system.

Essential revisions:

1) Throughout the study, the authors only examined 2-3 chemokine/cytokines that are known to have the SRSF1-binding sites in their 3' UTR. To strengthen the finding and the mechanism proposed, the authors should include analysis of an IRAK2-independent or a constitutively expressed cytokine/chemokine mRNA, and provide negative control data for key experiments including those in Figure 3 and Figure 4.

2) IRAK2 kinase activity is required for its interaction with RanBP2 and subsequent sumoylation modification. Can the authors further elaborate on this? Is IRAK2 autophosphorylation required for binding to RanBP2? If so, can the authors demonstrate this by using the phostag gel electrophoresis?

3) The exact function of IRAK2 binding to ALYREF is a little confusing. Data shown in Figure 4 suggest that the only function of IRAK2 is to remove SRSF1 from binding to the target mRNAs (therefore CXCL1 and TNF mRNAs were efficiently exported to the cytosol in SRSF1 knockdown cells in the IRAK2-independent manner). However, in subsequent presentation the authors also conclude that IRAK2-ALYREF interaction facilitates the assembly of nuclear export complex on target chemokine/cytokine mRNAs. Is there any inconsistency here?

4) In general, the RIP protocol used (low salt and without cross-linking) is sub-optimal. The lack of cross-linking means RNA and proteins re-association could easily occur post lysis, which, combined with the low-salt condition, means that some of the effects could be non-specific. Whilst it is not necessary for the authors to repeat all those experiments, such caveats regarding the RIP data should at least be noted and discussed in the text.

---

## [Author Response]

Essential revisions:1) Throughout the study, the authors only examined 2-3 chemokine/cytokines that are known to have the SRSF1-binding sites in their 3' UTR. To strengthen the finding and the mechanism proposed, the authors should include analysis of an IRAK2-independent or a constitutively expressed cytokine/chemokine mRNA, and provide negative control data for key experiments including those in Figure 3 and Figure 4.

This is a very good suggestion. The expression of *Ctnnb1* (β-catenin), a known target of SRSF1, is not regulated by IRAK2. Thus, we have now included *Ctnnb1* as a negative control in the RIP experiments (Figure 3 and Figure 4). The interactions of SRSF1/ALYREF with *Ctnnb1* mRNA were indeed not regulated by IRAK2 (Figure 3 and Figure 4). We have revised the Results subsections “IRAK2 mediates nuclear export of mRNAs of inflammatory genes” and “IRAK2 mediates the recruitment of ALYREF nuclear export factor to the target mRNAs”.

2) IRAK2 kinase activity is required for its interaction with RanBP2 and subsequent sumoylation modification. Can the authors further elaborate on this? Is IRAK2 autophosphorylation required for binding to RanBP2? If so, can the authors demonstrate this by using the phostag gel electrophoresis?

We appreciate this point. It has been reported that LPS stimulation induces the phosphorylation of IRAK2 at S136 and T140 (Weintz et al., 2010). We found that mutation at these two phosphorylation sites (IRAK2 S136A/T140A double mutant) abolished LPS-induced interaction of IRAK2 with RanBP2 and its nuclear localization (Figure 2—figure supplement 2), although the IRAK2 S136A/T140A double mutant still retained the interaction with importin-β. These findings suggest that the activation of IRAK2 may result in auto-phosphorylation at S136A and T140A, which in turn mediates the interaction with RanBP2. In support of this, it is indeed the modified form of IRAK2 was specifically co-immunoprecipitated by RanBP2 (Figure 2—figure supplement 3). Furthermore, the phos-tag gel electrophoresis showed that the phosphorylated IRAK2 preferentially binds to RanBp2 (Figure 2—figure supplement 3). We have revised the Results subsection” RanBP2-mediated IRAK2 sumoylation is required for its nuclear translocation”.

3) The exact function of IRAK2 binding to ALYREF is a little confusing. Data shown in Figure 4 suggest that the only function of IRAK2 is to remove SRSF1 from binding to the target mRNAs (therefore CXCL1 and TNF mRNAs were efficiently exported to the cytosol in SRSF1 knockdown cells in the IRAK2-independent manner). However, in subsequent presentation the authors also conclude that IRAK2-ALYREF interaction facilitates the assembly of nuclear export complex on target chemokine/cytokine mRNAs. Is there any inconsistency here?

IRAK2 has two functions in the nucleus: mediating the removal of SRSF1 and facilitating the assembly of nuclear export machinery via its interaction with ALYREF. SRSF1 prevents mRNA nuclear export by binding to the target mRNAs. IRAK2 phosphorylates SRSF1 in the nucleus, promoting the dissociation of SRSF1 from the target mRNAs (Figure 3). On the other hand, IRAK2 mediates the recruitment of nuclear export complex (Nxf1) to the target mRNAs through the interaction with ALYREF (Figure 4 and Figure 5). Based on these results, we proposed that the IRAK2-mediated dissociation of SRSF1 from the target mRNAs is required for the subsequent IRAK2-dependent recruitment of ALYREF/Nxf1 complex. In support of this, the interaction of IRAK2 to SRSF1 and to ALYREF are mutually exclusive (Figure 4).

4) In general, the RIP protocol used (low salt and without cross-linking) is sub-optimal. The lack of cross-linking means RNA and proteins re-association could easily occur post lysis, which, combined with the low-salt condition, means that some of the effects could be non-specific. Whilst it is not necessary for the authors to repeat all those experiments, such caveats regarding the RIP data should at least be noted and discussed in the text.

We appreciate the reviewers’ comments on this point. We have now repeated the key experiments using cross-linking method and data are included as supplemental data (Figure 3—figure supplement 4; and Figure 4—figure supplement 2).